# Siglec1-expressing subcapsular sinus macrophages provide soil for melanoma lymph node metastasis

Rohit Singh[1]*, Beom K Choi[2]*

[1]Division of Tumor Immunology, National Cancer Center, Goyang, Republic of Korea; [2]Biomedicine Production Branch, National Cancer Center, Goyang, Republic of Korea

**Abstract** Lymph nodes (LNs) are a common site of metastasis in solid cancers, and cutaneous melanomas show inherent properties of LN colonization. However, interactions between LN stroma and pioneer metastatic cells during metastatic colonization remain largely uncharacterized. Here we studied mice implanted with GFP-expressing melanoma cells to decipher early LN colonization events. We show that Siglec1-expressing subcapsular sinus (SCS) macrophages provide anchorage to pioneer metastatic cells. We performed in vitro co-culture to demonstrate that interactions between hypersialylated cancer cells and Siglec1 drive the proliferation of cancer cells. When comparing the transcriptome profile of Siglec1-interacting cancer cells against non-Siglec1-interacting cancer cells, we detected enrichment in positive regulators of cell cycle progression. Further, knockout of *St3gal3* sialyltransferase compromised the metastatic efficiency of tumor cells by reducing $\alpha-2,3$-linked sialylation. Thus, the interaction between Siglec1-expressing SCS macrophages and pioneer metastatic cells drives cell cycle progression and enables efficient metastatic colonization.

**\*For correspondence:**
11380@ncc.re.kr (BKC);
rohit@ncc.re.kr (RS)

**Competing interests:** The authors declare that no competing interests exist.

## Introduction

Lymphatic dissemination is a common metastasis route in solid cancers and LN metastasis is often associated with the severity of the disease (*Nathanson, 2003*). Although the clinical consequences of LN metastasis are debatable, recent reports have described clinically relevant events that take place during LN colonization by tumor cells (*Werner-Klein et al., 2018*; *Pereira et al., 2018*; *Brown et al., 2018*; *Lee et al., 2019*; *Naxerova et al., 2017*). Melanoma metastasizes to LNs very early in cancer progression and metastatic cells acquire driver genetic changes during colonization of LNs, suggesting the important role that LNs have in tumor evolution (*Werner-Klein et al., 2018*). These evolved post-LN-colonizing cancer cells with acquired colonization signatures show superior xenograft formation in mice compared with pre-colonizing cancer cells, suggesting a possible reason as to why metastasis-positive LNs are linked to disease severity in cancer (*Werner-Klein et al., 2018*). Moreover, LNs provide a foothold for metastatic cells to disseminate further. Besides lymphatic dissemination to downstream LNs, during LN colonization metastatic cells gain access to the blood vessels in LNs and take a hematogenous route to colonize peripheral organs (*Pereira et al., 2018*; *Brown et al., 2018*). Despite all clinically important roles that LNs have in the systemic spread of evolved cancer cells, our knowledge of the early events of LN colonization by pioneer metastatic cells is very limited (*Naxerova et al., 2017*).

The distal organ metastatic colonization process is organ-specific and requires interactions between disseminated tumor cells (DTCs; seed) and distal organs (soil) to reinitiate growth (*Peinado et al., 2017*; *Massagué and Obenauf, 2016*). Although primary tumors release many early DTCs into the circulation, which are the precursors of metastasis, very few eventually form distal

**eLife digest** Cancer cells can leave the site where they arise and travel to other organs. Very few of the cancer cells that make this journey will survive long enough to form new tumors (also known as 'metastases'). However, melanoma cells – the most aggressive type of skin cancer cells – are an exception. These cells will often colonize their nearest lymph nodes and melanoma patients with metastases in the lymph nodes are less likely to survive than those patients without them.

Previous studies have shown that melanoma cells arrive at a lymph node and first proliferate in the region at the edge of this organ, known as the subcapsular sinus, before moving to the center. However, it was not understood how melanoma cells manage to survive in the subcapsular sinus.

Now, Singh and Choi have tracked fluorescent melanoma cells to observe how they interact with the cells in the lymph nodes in mice. Melanoma cells have 'sticky' proteins coated with sugars on their surface. The results show that when the cells arrive in the subcapsular sinus these proteins bind to a receptor called Siglec1 located on the surface of immune cells called macrophages, which are also present. In this way, the melanoma cells anchor themselves in the lymph node. Moreover, binding Siglec1 helps melanoma cells survive and proliferate. In a last set of experiments, Singh and Choi deleted the enzyme responsible for making the sugar molecules in melanoma cells. Without the sugar coat, melanoma cells were less able to anchor themselves and grow within the mouse lymph nodes.

Lymph nodes are often the first stop for melanoma cells on the way to other organs. Therefore, understanding the interaction between melanoma cells and macrophages might be useful for developing therapies that could disrupt this process and treat this aggressive cancer.

metastasis. The majority of DTCs that land in unfamiliar distal sites fail to survive the hostile microenvironment of the new-found stroma. This inefficiency of metastasis suggests that distal organ colonization is a bottleneck of the metastasis colonization process. To gain a foothold in new landing sites, pioneer metastatic cells require active participation of the host stroma to obtain survival and viability signals (*Mehlen and Puisieux, 2006*; *Buchheit et al., 2014*). A better understanding of the events that define this interaction between metastatic cells and host organ stroma is needed to identify actionable vulnerabilities of the metastatic colonization process. The current paradigm posits that LN metastatic colonization begins with the arrest of metastatic cells in the LN subcapsular sinus (SCS) (*Das et al., 2013*; *Jeong et al., 2015*). However, in LN metastasis, focus is often given to post-metastatic colonization events, whereas the interactions between metastatic cells and host stroma that leads to LN colonization by tumor cells remains largely uncharacterized (*Nathanson, 2003*; *Lee et al., 2019*; *Das et al., 2013*; *Jeong et al., 2015*; *Olmeda et al., 2017*). Therefore, using enhanced green fluorescent protein (GFP)-expressing mouse melanoma cells, we investigated the early LN metastatic colonization events and how LN supports the newly arrived DTCs.

## Results

### LN metastatic colonization begins with interactions between pioneer metastatic cells and Siglec1[+] SCS macrophages

To decipher the earliest metastatic events in LN, we first defined the stage at which melanoma LN metastasis formation takes place. Enhanced green fluorescent protein-expressing B16F10 (B16-GFP) melanoma cells were injected into mouse footpads to form primary tumor (*Figure 1—figure supplement 1A*). The advantage of this model is that it mimics the natural course of metastasis, that is, primary tumor growth, pre-metastatic conditioning of draining LNs, and dissemination of tumor cells followed by LN colonization (*Jeong et al., 2015*; *Harrell et al., 2007*), while GFP enables us to track cancer cells even as single cells. Pioneer metastatic cells could be observed in the SCS of sentinel popliteal LNs lined by Lyve-1[+] lymphatic endothelial cells as early as 2 weeks after inoculation and before the appearance of any visible macroscopic metastatic foci (*Figure 1—figure supplement 1B*). Given the cell composition of the LN SCS and prominent presence of SCS macrophages in the sinus (*Gray and Cyster, 2012*), we reasoned that pioneer metastatic cells may use SCS macrophages to find an initial anchorage point. Therefore, we stained the LN sections positive for B16-GFP cells

with anti-Siglec1 (CD169) antibody, the surface protein highly expressed on macrophages on the floor of the SCS and in the LN medullary region (*Gray and Cyster, 2012*; *Nakamura et al., 2002*). Of note, by depleting LN-resident macrophages through injecting clodronate liposomes into the mouse footpads, we confirmed that Siglec1 is primarily expressed on SCS and medullary sinus macrophages in LNs (*Figure 1—figure supplement 2*). We found the pioneer metastatic cells in close contact with SCS macrophages within the LN SCS, suggesting they interact with each other right after metastatic cells enter the SCS (*Figure 1A and B*; *Video 1*). Likewise, we confirmed this SCS macrophage–tumor cell interaction using another cell surface marker expressed on SCS macrophages, CD11b (*Figure 1—figure supplement 1C*). We analyzed the sections from day 17 to day 21, and found that after initial cancer cell attachment to SCS macrophages, small metastatic foci developed within the SCS (*Figure 1C*). With the increase in size, the foci exerted pressure on the SCS macrophage–lymphatic lining, which eventually lead to disruption of the sinus lining and metastatic cells entering the LN cortex (*Figure 1D*). Of note, to test the generalizability of our observations, we implanted an EGFP-expressing 4T1 mouse breast cancer cell line into mouse footpads and found that breast cancer cells also used SCS macrophages to find a foothold in LNs (*Figure 1—figure supplement 3*). Taken together with previous observations (*Das et al., 2013*; *Jeong et al., 2015*), where LN metastatic cells from different tumor types remain in the SCS and form a metastatic focus before invading into the LN cortex, our observations indicate that other tumors metastasizing to LNs may also use a LN colonization mechanism similar to melanoma.

## Siglec1 binds to α2,3-sialylated proteins on cancer cells

Increased protein sialylation is characteristic of several cancer types including melanomas (*Kim and Varki, 1997*; *Agrawal et al., 2017*). Using *Maackia amurensis* II and *Sambucus nigra* lectins (α−2,3- and α−2,6-sialylation-specific, respectively) (*Varki, 2007*) we found more than 10-fold higher cell surface α2,3- and α2,6-linked sialylation in B16F10 melanoma cells compared with non-tumorigenic mouse melanocytes, melan-A cell line (*Figure 1—figure supplement 4A–D*) (*Bennett et al., 1987*). Of note, melan-A cells showed only basal levels of α2,3-sialylation. As Siglec1 is a sialic-acid-recognizing protein that interacts with α2,3- and α2,6-linked sialylated proteins (*Crocker et al., 1999*), we hypothesized that it is Siglec1 itself that is responsible for the SCS macrophage–tumor cell interaction (*Nath et al., 1999*). To confirm this, we used cell adhesion assay. HEK293T cells were grown as a monolayer in chamber slides and transfected with plasmid expressing mouse *Siglec1* or empty plasmid as a mock control. We found significantly higher adherence of B16-GFP cells to Siglec1-expressing HEK293T cells compared with mock-transfected cells (*Figure 1E,F*). Additionally, we confirmed Siglec1 binding to cancer cells by flow cytometry and microscopy employing recombinant mSiglec1(ECD)-mFC protein (*Figure 1—figure supplement 4E,F*). Furthermore, treatment with sialidase abolished the Siglec1 binding to cancer cells while removing the α2,3- sialylation completely and α2,6- sialylation by one third, suggesting Siglec1 preferentially binds to α2,3-sialylated protein on cancer cells (*Figure 1—figure supplement 5A–F*). Based on the above in vivo and in vitro results, we suggest that Siglec1-expressing SCS macrophages provide the 'soil' for incoming metastatic cells and prevent their washout with lymph flow.

## Siglec1-interacting cancer cells show higher proliferation

With limited chances of survival, disseminated tumor cells (DTCs) land in distal metastasis sites (*Massagué and Obenauf, 2016*). To establish a new colony, pioneer metastatic cells must overcome anoikis and amorphosis, and resume proliferation by establishing adhesive and signaling interactions with host tissue (*Mehlen and Puisieux, 2006*; *Buchheit et al., 2014*). As our data confirmed cell-to-cell contact between SCS macrophages and pioneer metastatic cells, we investigated the functional consequences of this interaction for tumor cells. We used apoptosis marker cleaved-caspase-3 and cell proliferation marker Ki67 to determine the proliferation status of pioneer metastatic cells and found that they were in a non-proliferative non-apoptotic state when they landed in the LN SCS, and resumed proliferation after they came into contact with SCS macrophages (*Figure 2A–D*). To confirm this, we used an in vitro co-culture method. Mouse Siglec1 was transiently expressed in HEK293T cells and B16-GFP cells were co-cultured with these Siglec1-expressing cells (hereafter referred to as HEK[Siglec1] and HEK[Mock] for cells transfected with plasmid expressing mouse *Siglec1* and empty plasmid, respectively; *Figure 2E*, *Figure 2—figure supplement 1A*). Of note, Siglec1

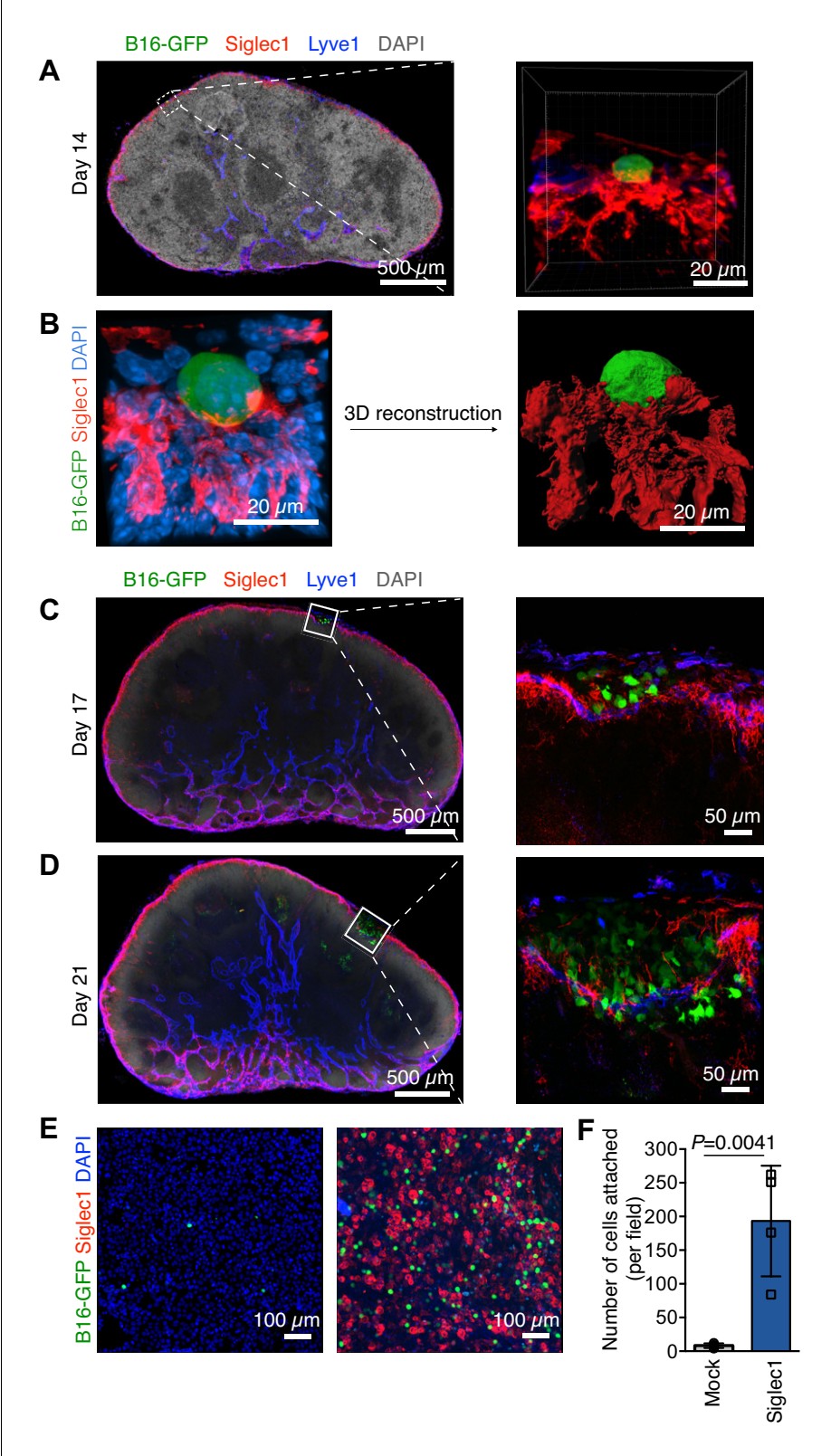

**Figure 1.** Steps of lymph node (LN) metastatic colonization. B6 mice were implanted with B16-GFP cancer cells in footpads. Representative laser scanning microscopy images of cryosectioned sentinel popliteal LNs (pLNs) on indicated days are shown. (**A**) Image taken at 14 days post-inoculation; pioneer metastatic cells (green) were evident in the subcapsular sinus (SCS) of pLN. Here, metastatic cell could be seen in close contact with Siglec1[+] SCS macrophages (red) in the LN SCS lined by lymphatic endothelial cells (blue). (**B**) A laser scanning microscope image of early metastatic B16F10 cell

*Figure 1 continued on next page*

*Figure 1 continued*

(green) in close contact with LN SCS macrophages (Siglec1, red; nucleus, blue) and its 3D reconstructed image. (C) Metastatic cells resumed proliferation and formed microscopic metastatic foci within the LN SCS. (D) Metastatic foci grew and disrupted the SCS macrophage–lymphatic endothelial cell lining and entered the LN cortex. Data representative of four biologically independent experiments. (E, F) Adhesion assays of B16-GFP cells on monolayers of HEK293T or Siglec-1 expressing HEK293T cells. (E) Representative photomicrographs of adherent B16-GFP cells on the monolayer of indicated HEK293T cells. (F) Quantification of the number of adherent B16-GFP cells on monolayers. Data are ±s.d.; n = 4 biologically independent experiments. *P*-value was calculated by two-tailed, unpaired *t*-test.

The online version of this article includes the following source data and figure supplement(s) for figure 1:

**Source data 1.** This spreadsheet contains the source data for *Figure 1F*.
**Figure supplement 1.** Approach for visualization of pioneer metastatic cells and early metastatic events in LNs.
**Figure supplement 2.** Siglec1 is expressed on the macrophages on the floor of the LN SCS and medullary region.
**Figure supplement 3.** 4T1 mouse breast cancer cells use LN SCS macrophages to find footholds during LN metastasis.
**Figure supplement 4.** Siglec1 binds to hypersialylated cell surface proteins of mouse melanoma cells.
**Figure supplement 4—source data 1.** This spreadsheet contains the source data for figure supplement 4.
**Figure supplement 5.** Siglec1 binds to mouse melanoma cells in an α−2,3 sialylation-specific manner.
**Figure supplement 5—source data 1.** This spreadsheet contains the source data for figure supplement 5.

expression did not affect the proliferation status of HEK293T cells (*Figure 2—figure supplement 1B*). To mimic the non-adherent nature of DTCs, we co-cultured cells in low binding culture plates. Of note, we observed that cancer cells cultured in non-adherent conditions showed significantly reduced Ki67 expression, similar to DTCs (*Figure 2—figure supplement 1C,D*) (*Pantel and Brakenhoff, 2004*). We found more Ki67[+] proliferating B16-GFP cells in tumor cells-HEK[Siglec1] co-cultures compared with cells co-cultured with HEK[Mock] (*Figure 2F,G*). To exclude the possibility that expression of Siglec1 may induce HEK293T cells to secrete some pro-proliferation factors and in turn induce B16-GFP cell proliferation, we prepared conditioned media from HEK[Mock] and HEK[Siglec1] cells and treated the B16-GFP cells in a similar setting as the co-culture experiments. Conditioned media did not affect the proliferation of B16-GFP cells (*Figure 2—figure supplement 1E,F*). We next tested the role of Siglec1 in the survival of cancer cells. We assessed Annexin V staining in B16-GFP cells after co-culturing B16-GFP cells with HEK[Mock] and HEK[Siglec1] cells. B16-GFP cells co-cultured with HEK[Siglec1] cells had significantly lower Annexin V-positive apoptotic cells (*Figure 2H,I*). We further validated these results using a mouse macrophage cell line, J774A.1, which expresses Siglec1 (*Figure 2—figure supplement 2A*). We used siRNA to knockdown *Siglec1* expression in J774A.1 cells and performed co-culture experiments with B16-GFP cells (*Figure 2—figure supplement 2B–D*). We observed decreased proliferation of B16-GFP cells co-cultured with these *Siglec1* knockdown J774A.1 macrophages compared with B16-GFP cells co-cultured with control siRNA-transfected macrophages (*Figure 2—figure supplement 2E, F*). Furthermore, we observed a significant increase in Annexin V-positive apoptotic cancer cells when they were co-cultured with *Siglec1*-knocked down macrophages (*Figure 2—figure supplement 2G,H*). Taken together, our data revealed a role for Siglec1 in cancer cell survival and proliferation during the initial metastatic colonization of LNs by pioneer metastatic cells.

## Siglec1-interacting cancer cells show higher phosphorylation of AKT and ERK1/2

Successful metastatic colonization requires activation of the pro-survival PI3K/AKT and ERK pathways (*Buchheit et al., 2014*). Because these two cascades are crucial for metastatic colonization, we tested whether they were sensitive to Siglec1-dependent cancer cell interactions. After

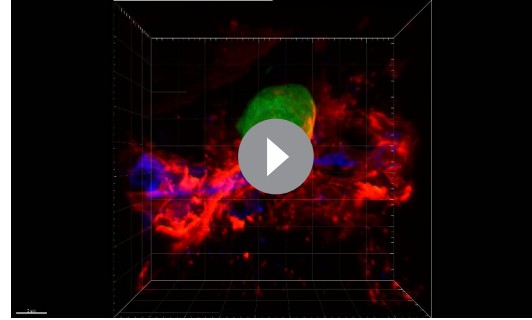

**Video 1.** A close interaction of SCS macrophages (red) and metastatic cell (green) visualized in 3D space within lymph node SCS lined by lymphatic endothelial cells (blue). We can see the SCS macrophage protrusion which spreads over wide surface of cancer cells, this contact provides anchorage to newly arrived pioneer metastatic cell in SCS (nuclei, white).
https://elifesciences.org/articles/48916#video1

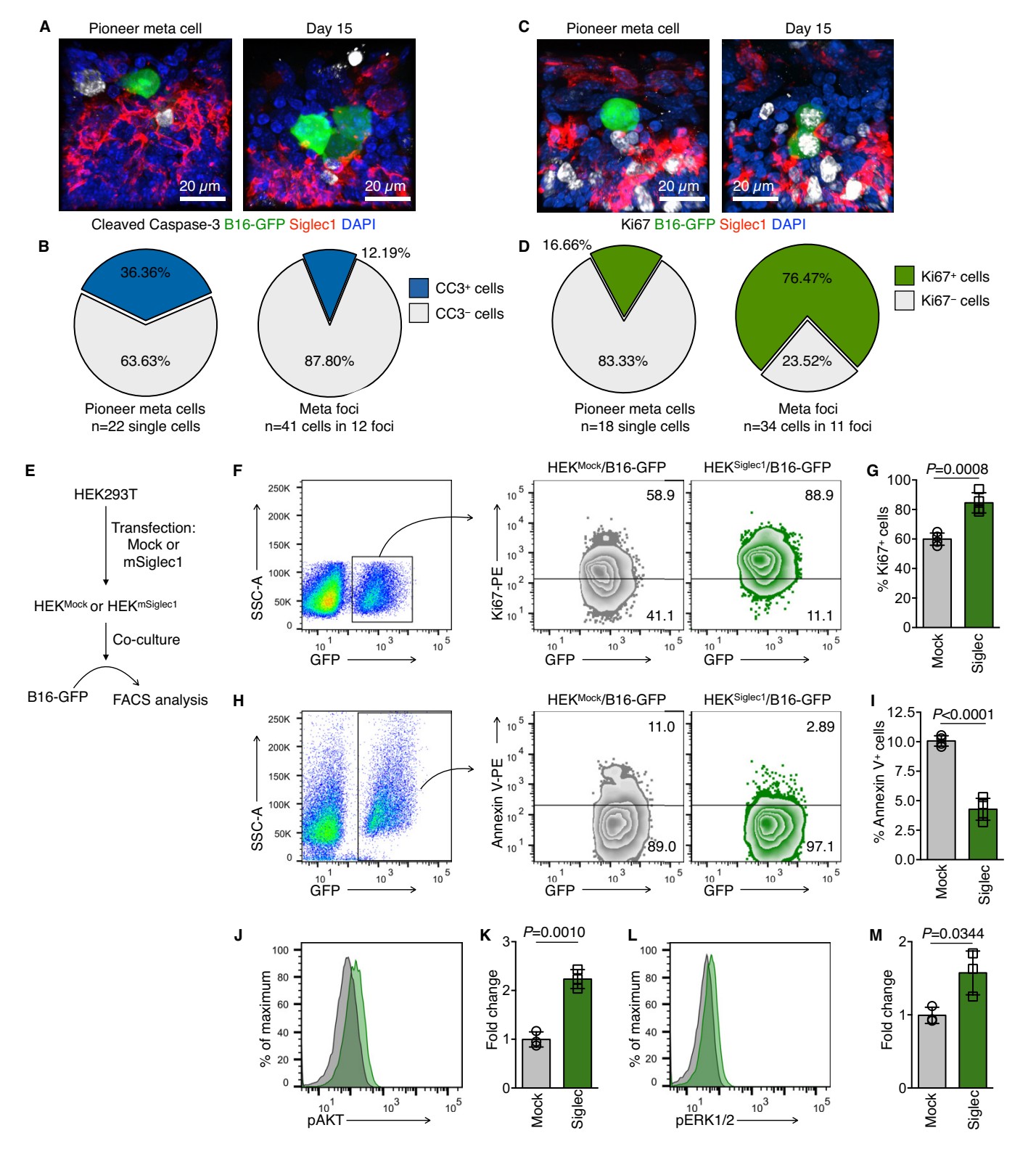

**Figure 2.** Siglec1 on SCS macrophages provides growth support to metastatic cells. (A–D) Apoptosis (cleaved caspase-3, white; **A, B**) and proliferation (Ki67, white; **C, D**) in newly deposited metastatic cells in the LN SCS. Pioneer metastatic cells were in a non-apoptotic non-proliferative state when they landed in the SCS and resumed growth after arrest by SCS macrophages. Data representative of four biologically independent experiments. (**E**) Schematic of co-culture experimental procedure. HEK293T cells were transfected with empty mammalian expression plasmid or mSiglec1-expressing

*Figure 2 continued on next page*

*Figure 2 continued*

plasmid. Cells were cultured for 3 days before use in co-culture experiments with B16-GFP cells. (F, G) Proliferation of B16-GFP cells measured by Ki67 expression after 18 hr co-culture with HEK^Mock or HEK^Siglec1 cells (n = 4 independent experiments, *P*-value by two-tailed, unpaired *t*-test). (H, I) Apoptosis in B16-GFP cells after 18 hr co-culture with HEK^Mock or HEK^Siglec1 cells measured by Annexin V staining (n = 4 independent experiments, *P*-value by two-tailed, unpaired *t*-test). (J–M) Intracellular Phosflow staining to access phosphorylated AKT (pAKT; J, K) and phosphorylated ERK1/2 (pERK; L, M) levels in B16-GFP cells after 18 hr co-culture with HEK^Mock or HEK^Siglec1 cells. Bar graphs represent fold change in phosphorylation over B16-GFP co-cultured with HEK^Mock (n = 3 independent experiments; *P*-value was calculated by two-tailed, unpaired *t*-test). CC3, cleaved caspase-3; meta, metastasis.

The online version of this article includes the following source data and figure supplement(s) for figure 2:

**Source data 1.** This spreadsheet contains the source data for *Figure 2*.
**Figure supplement 1.** HEK^Mock and HEK^Siglec1 cells.
**Figure supplement 1—source data 1.** This spreadsheet contains the source data for figure supplement 1.
**Figure supplement 2.** Knockdown of *Siglec1* in J774A.1 mouse macrophages and subculture with B16-GFP cells.
**Figure supplement 2—source data 1.** This spreadsheet contains the source data for figure supplement 2.

co-culturing B16-GFP cells with HEK^Siglec1 or HEK^Mock for 18 hr, we performed intracellular Phosflow analysis. Siglec1 engagement with cancer cells induced AKT phosphorylation at S473, which is a PI3K-mediated AKT-activating event (*Figure 2J,K*). Furthermore, in a similar setting, we found higher levels of phosphorylated ERK1/2 (ERK1, T203/Y205; ERK2, T183/Y185; *Figure 2L,M*) in cancer cells co-cultured with HEK^Siglec1. We further analyzed AKT and ERK1/2 phosphorylation in B16-GFP cells co-cultured with *Siglec1*-knocked down J774A.1 cells. Consistent with the reduced proliferation and increased apoptosis in B16-GFP co-cultured with *Siglec1*-knocked down J774A.1 cells, we found reduced AKT and ERK1/2 phosphorylation in comparison with cells co-cultured with J774A.1 cells transfected with control siRNA (*Figure 2—figure supplement 2I–L*). Collectively, these results suggest that Siglec1 interactions provide growth cues to cancer cells. Therefore, these interactions may facilitate the LN colonization of melanoma cells.

## Siglec1 drives transcriptional programs necessary for metastatic colonization

Next, to determine the molecular outcome of tumor cell and Siglec1 interactions, we performed a whole transcriptome analysis (*Figure 3A*). Compared with tumor cells that were not interacting with Siglec1, tumor cells cultured with HEK^Siglec1 showed differential expression (fold change >1.5; raw *P*<0.05) of 239 genes, of which 68 were upregulated and 171 were downregulated (*Figure 3B*, *Figure 3—source data 1*). A pathway analysis of the upregulated genes revealed that tumor cells from HEK^Siglec1–tumor cell co-culture displayed enrichment in positive regulators of cell cycle and DNA replication compared with cancer cells cultured with mock-transfected HEK293T cells (*Figure 3C*). Furthermore, gene ontology analysis revealed that upregulated genes were mainly involved in cell cycle-related biological processes (*Figure 3D,E*). Consistent with FACS analysis of Ki67 staining, RNA sequencing results confirmed higher levels of Ki67 expression in HEK^Siglec1 co-cultured cancer cells (*Figure 3E*, *Figure 3—source data 1*). Additionally, we observed downregulation of apoptosis-related (*Bbc3*, *Rras*, *Ddit3*, *Ddit4*) and tumor suppressor (*Hes1*, *Ahnak*, *Timp3*, *Smad7*, *Ndrg1*) genes in Siglec1-interacting cancer cells (*Figure 3—source data 1*). Thus, the interaction between Siglec1 and cancer cells drove cell cycle progression and proliferation in cancer cells. In light of the prior knowledge of the vulnerabilities of metastatic processes, Siglec1-mediated interactions between SCS macrophages and metastatic cells may provide the vital support required to overcome dormancy and initiate successful LN colonization (*Pantel and Brakenhoff, 2004*; *Sosa et al., 2014*).

## Siglec1-interacting cancer cells show high PLK1 activation

As we observed a shift towards increased cell cycle activity and proliferation in Siglec1-interacting cancer cells, we sought to further explain the mechanism by which Siglec1 interaction induces cell cycle activation in tumor cells by examining the critical mitotic events. To this end, we revisited our RNA sequencing results to identify important molecular events that regulate mitotic entry. Gene expression data showed significant upregulation in cell cycle master regulator PLK1 expression in co-culture experiments (*Figure 3E*, *Figure 3—source data 1*) (*Paschal et al., 2012*). This prompted us to compare PLK1 activation in primary tumor and early LN metastatic cells. We used Phosflow to

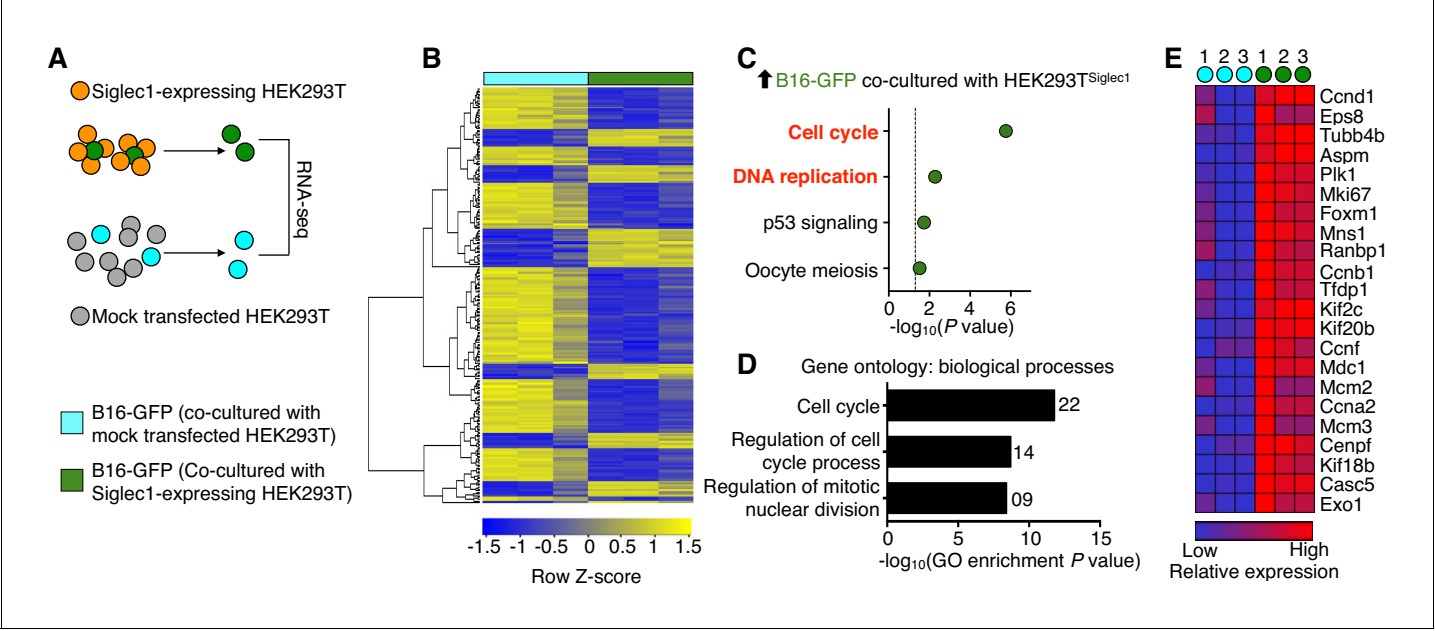

**Figure 3.** Siglec1 cancer cell interaction drives cell cycle progression. (**A**) Schematic of the experimental design. B16-GFP cells co-cultured with Siglec1-expressing and mock transfected HEK293T cells were FACS sorted and processed for RNA sequencing. (**B**) Heat map of two-way hierarchical clustering of differentially expressed genes satisfying FC>1.5 and raw *P*<0.05 using Z-score for normalized log2 values. (**C**) KEGG pathways over-represented (*P*<0.05) among upregulated genes in cancer cells in B16-GFP and HEK $^{Siglec1}$ co-cultured cells. (**D, E**) Siglec1-interacting cancer cells showing transcriptional induction of cell cycle. Shown are the top three bioprocess gene ontology terms in upregulated genes (**D**) and heat map of upregulated cell cycle genes (**E**). Differentially expressed genes are listed in *Figure 3—source data 1* (data is from three independent experiments).
The online version of this article includes the following source data for figure 3:

**Source data 1.** This spreadsheet contains the list of differentially expressed genes.

compare PLK1 phosphorylation in primary tumor and LN metastatic cells. PLK1 phosphorylation at T210 is a mitosis-specific G2/M transition-activating event (*Paschal et al., 2012*). We found 1.8-fold higher activation of PLK1 in LN metastatic cells in comparison with primary tumor cells (*Figure 4A, B*). Consistently, in in vitro experiments, we found marked increase in mitosis-committed cancer cells when B16-GFP cells were co-cultured with HEK$^{Siglec1}$ in comparison with HEK$^{Mock}$ (*Figure 4C,D*). With a reverse approach, where we knocked down *Siglec1* using siRNA in J774A.1 macrophages and performed co-culture experiments with B16-GFP cells, we found that knockdown of *Siglec1* significantly decreased PLK1 phosphorylation (*Figure 4E–G*, *Figure 2—figure supplement 2A–D*). Taken together, Siglec1-interacting tumor cells showed higher PLK1 phosphorylation than those that did not interact with Siglec1. This strong mitotic induction observed in Siglec1-interacting tumor cells and confirmed by high PLK1 phosphorylation during early LN metastatic colonization supports our model whereby Siglec1 confers a growth advantage to pioneer metastatic cells. This presence of a highly growth permissive 'soil' reflects the high prevalence of LN metastasis in melanoma during the early stages of primary tumor growth, even before metastasis to peripheral organs.

## Role of α2,3-sialylation in LN metastasis

Hypersialylation in cancers, including melanomas, is directly linked to the metastatic potential of cancer cells (*Agrawal et al., 2017*; *Bos et al., 2009*; *Schultz et al., 2012*). In previous experiments, we established that Siglec1 preferentially binds to $\alpha-2,3$-linked sialylation (*Figure 1—figure supplement 5A–F*). One sialyltransferase, ST3 beta-galactoside alpha-2,3-sialyltransferase 3 (St3gal3) is known to produce Siglec-binding $\alpha-2,3$ sialylation in mice and is linked to metastasis and poor prognosis in several cancer types (*Guo et al., 2011*; *Pérez-Garay et al., 2010*; *Cui et al., 2011*; *Chang et al., 2006*). Hence, we knocked out *St3gal3* in B16-GFP cells using CRISPR-mediated single base pair deletion (*Figure 5—figure supplement 1A,B*), which decreased $\alpha-2,3$ sialylation of cell

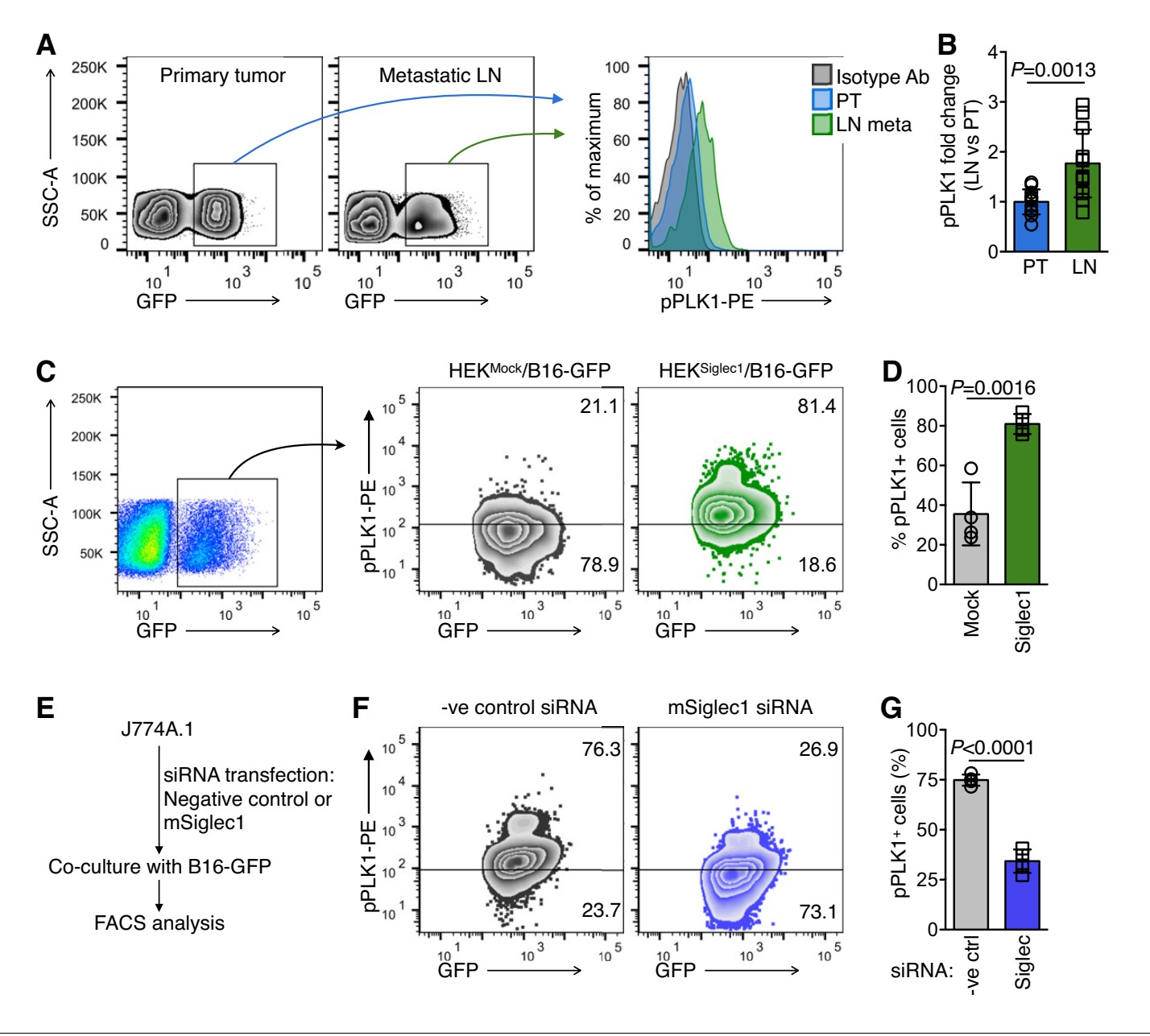

**Figure 4.** Siglec1-interacting cancer cells show high mitotic commitment. (**A**) FACS plots showing phosphorylated PLK1 (pPLK1) in primary tumor (PT) and LN metastatic cells (LN meta). (**B**) Fold change in pPLK1 in LN meta vs. PT cells calculated from A (n = 12 from three independent experiments with four animals per experiment; bar represents ±s.d., *P*-value was calculated by two-tailed, unpaired *t*-test). (**C, D**) PLK1 phosphorylation in B16-GFP cells after 18 hr co-culture with indicated cancer cells/HEK293T co-culture. Representative FACS plot (**C**) and quantification (**D**) are shown (n = 4 independent experiments). (**E**) Schema of siRNA transfection in J774A.1 mouse macrophage cells and co-culture with B16-GFP cells. (**F, G**) PLK1 phosphorylation in B16-GFP cells after 18 hr co-culture with *Siglec1*-knocked down and control J774A.1 cells.( **G**) Quantification of (**F**) (n = 4 independent experiments, *P*-value was calculated by two-tailed, unpaired *t*-test).
The online version of this article includes the following source data for figure 4:

**Source data 1.** This spreadsheet contains the source data for *Figure 4*.

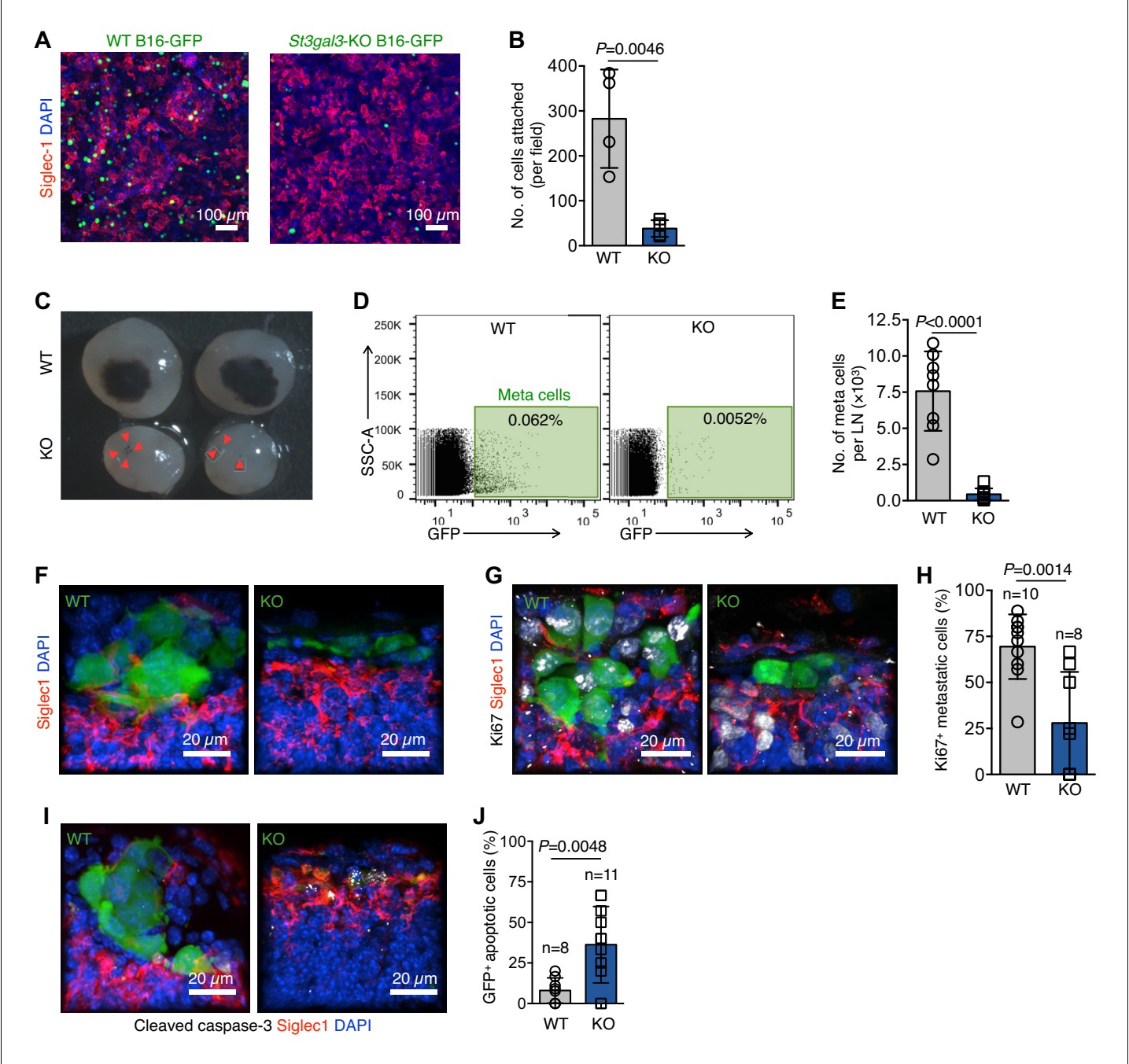

**Figure 5.** α−2,3-linked sialylation and melanoma LN metastasis. (**A**) In vitro adhesion assay of wildtype (WT) and *St3gal3*-knockout (KO) B16-GFP cell lines to a Siglec1-expressing HEK293T monolayer. (**B**) The numbers of cells attached to the monolayer are plotted. Data are ±s.d.; n = 4 biologically independent experiments. *P*-value by two-tailed, unpaired *t*-test. (**C**) Representative images of metastasis bearing LNs from WT or KO B16-GFP tumor cells implanted in mice 3 weeks after injection. Red arrowheads indicate metastatic foci. (**D**) Representative plots of GFP+ metastasized cells in LNs of mice bearing 3 week primary WT or *St3gal3* KO B16-GFP melanoma tumors. (**E**) Graph showing total metastasis burden per LN in mice implanted with WT or *St3gal3* KO B16-GFP (quantification of C; n = 8 samples per group; *P*-value was calculated by two-tailed, unpaired *t*-test). (**F**) Morphological features of early metastatic foci (day 17) in the LN SCS of mice injected with WT or *St3gal3* KO B16-GFP (green, B16-GFP; red, SCS macrophages). (**G**) Representative confocal images of Ki67+ (white) nuclei in metastatic foci formed by WT or *St3gal3* KO B16-GFP cells in the LN SCS. (**H**) Quantification of Ki67+ nuclei in metastatic foci in (**G**). (**I**) Representative confocal image of apoptosis in WT and *St3gal3* KO B16-GFP cells during the early stages of LN metastatic colonization measured by cleaved caspase-3+ (white) staining. (**J**) Quantification of cleaved caspase-3+ cells in (**I**). Data are from a total of six mice in each group from two independent experiments; *P*-value by two-tailed, unpaired *t*-test.

The online version of this article includes the following source data and figure supplement(s) for figure 5:

*Figure 5 continued on next page*

*Figure 5 continued*

**Source data 1.** This spreadsheet contains the source data for *Figure 5*.
**Figure supplement 1.** ST3 beta-galactoside alpha-2,3-sialyltransferase 3 (*St3gal3*) knockout cell line.
**Figure supplement 1—source data 1.** This spreadsheet contains the source data for figure supplement 1.
**Figure supplement 2.** Primary tumor lymphatic vessels do not express Siglec1.
**Figure supplement 3.** Representative image of metastasis in popliteal LNs of wild type (WT) and *St3gal3* knockout (KO) B16-GFP primary tumor-bearing mice.

surface proteins by 20% (*Figure 5—figure supplement 1C,D*). Additionally, *St3gal3* knockout (KO) cells showed approximately 70% reduced cell surface binding of Siglec1 (*Figure 5—figure supplement 1E,F*). Of note, *St3gal3* KO did not affect the proliferation of B16-GFP cells in culture, in vitro adhesion and migration, or growth as primary tumors (*Figure 5—figure supplement 1G–K*), but it abrogated the ability of B16-GFP to adhere to a Siglec1-expressing HEK293T cell monolayer (*Figure 5A,B*). Notably, *St3gal3* KO severely reduced the LN metastasis burden (*Figure 5C–E*). To explore the possibility of Siglec1 expression on primary tumor lymphatic vessels, which might have affected the initial egress of cancer cells from the primary tumor, we searched for Siglec1 expression on primary tumor lymphatic vessels and found none (*Figure 5—figure supplement 2*). Next, we sought to explain the reduced metastasis by comparing the morphological features of early metastatic foci and quantifying the proliferation and survival of metastatic cells in the LN SCS. Immunohistological analysis at an early metastasis stage (day 17) revealed that although *St3gal3* KO B16-GFP cells are present in the LN SCS, they did not form foci and were not in contact with SCS macrophages (*Figure 5F*). Whereas, WT B16-GFP cells formed tight-knit foci in contact with SCS macrophages (*Figure 5F*). We further investigated whether this deprivation of Siglec1-derived signals from SCS macrophages affected the proliferation and survival of *St3gal3* KO metastatic cells during early colonization. Ki67 staining revealed that *St3gal3* KO B16-GFP cells showed severely reduced proliferation in the SCS in comparison with WT B16-GFP cells (*Figure 5G,H*). Furthermore, early metastatic cells originating from *St3gal3* KO cells showed significantly decreased survival in the SCS (*Figure 5I,J*). Three weeks after tumor cell implantation, a large metastatic focus had formed in the LN, which disrupted the SCS macrophages lining the LN sinus, but *St3gal3* KO tumors only formed small cell clusters, which crossed the SCS without growing in size in the SCS, suggesting a lack of ability to adhere to the SCS macrophages and hence reduced metastatic growth (*Figure 5—figure supplement 3*). Thus, overall $\alpha-2,3$-sialylation abundance on tumor cells affects the efficiency of LN metastasis formation. Combined, these results suggest that the survival and proliferation of pioneer metastatic cells in the LN SCS is crucial process of metastatic colonization, and abrogation of SCS macrophage–tumor cell binding slowed down the metastatic colonization considerably.

## Discussion

Melanoma has a predilection for metastasizing to LNs in the early stages of the disease (*Bedrosian et al., 2000*). The high prevalence of LN metastasis early during tumor progression suggests the presence of a metastasis-permissive environment in LNs. Our data provide a possible explanation for this. We propose a model whereby lymphatic disseminated metastatic cells directly interact with SCS macrophages just after landing in the LN SCS. The overall abundance and continuous presence of Siglec1[+] macrophages in the SCS facilitates efficient metastatic seeding by providing adherence and promoting cell cycle progression in otherwise vulnerable DTCs. Our model is in agreement with the sequence of events leading to metastatic colonization of LN observed by Das et al. and Jeong et al. using in vivo imaging, whereby they observed that pioneer metastatic cells remain in the LN SCS to form metastatic foci that grow in size and disrupts the SCS to invade the LN cortex (*Das et al., 2013*; *Jeong et al., 2015*).

LN SCS macrophages have been implicated in anti-tumor responses (*Asano et al., 2011*; *Pucci et al., 2016*). We show here another facet of Siglec1[+] SCS macrophages where they physically interact with hypersialylated metastatic cells, conferring growth advantages upon them. Genes affected by Siglec1-cancer cell interactions were not only involved in cell cycle progression and DNA replication, but also in apoptosis and tumor suppression. Accordingly, reduced sialylation in tumor cells reduced the efficiency of metastasis. Our findings also suggest how the organ-specific LN

metastasis trait of melanoma arises, and countering hypersialylation may therefore prove to be therapeutically beneficial in preventing LN metastasis and subsequent spread to distal organs.

# Materials and methods

## Key resources table

| Reagent type (species) or resource | Designation | Source or reference | Identifiers | Additional information |
|---|---|---|---|---|
| Cell line (mouse) | B16F10 | ATCC | Cat# CRL-6475, RRID:CVCL_0159 | |
| Cell line (mouse) | 4T1 | ATCC | Cat# CRL-2539 RRID:CVCL_0125 | |
| Cell line (human) | HEK293T | ATCC | Cat# CRL-3216, RRID:CVCL_0063 | |
| Cell line (mouse) | J774A.1 | ATCC | Cat# TIB-67, RRID:CVCL_0358 | |
| Antibody | anti-mouse CD169 (Siglec1) (clone #3D6.112) | Bio-Rad | Cat# MCA884, RRID:AB_322416 | (1:200) |
| Antibody | Rabbit anti-mouse LYVE-1 antibody | Angiobio | Cat# 11–034, RRID:AB_2813732 | (1:200) |
| Antibody | Rabbit anti-Ki67 antibody | Abcam | Cat# ab15580, RRID:AB_443209 | (1:200) |
| Antibody | Rabbit anti-cleaved Caspase-3 (Asp175) antibody | Cell Signaling Technology | Cat# 9579, RRID:AB_10897512 | (1:200) |
| Antibody | Rabbit Anti-GFP antibody | Abcam | Cat# ab6556, RRID:AB_305564 | (1:200) |
| Antibody | Cy3 AffiniPure Goat Anti-Rat IgG (H+L) | Jackson Immuno Research | Cat# 112-165-167, RRID:AB_2338251 | (1:500) |
| Antibody | Cy5 AffiniPure Goat Anti-Rabbit IgG (H+L) | Jackson Immuno Research | Cat# 111-175-144, RRID:AB_2338013 | (1:500) |
| Antibody | PE mouse anti-Ki67 antibody (Clone # B65) | BD Biosciences | Cat# 556027, RRID:AB_2266296 | As recommended by manufacturer |
| Antibody | Alexa Fluor647-anti-mouse CD169 (Siglec1) antibody (clone #3D6.112) | BioLegend | Cat# 142408, RRID:AB_2563621 | As recommended by manufacturer |
| Antibody | PE mouse IgG1 Isotype control antibody (Clone# MOPC-21) | BD Biosciences | Cat# 556027, RRID:AB_2266296 | As recommended by manufacturer |
| Antibody | BD Phosflow PE mouse anti-Akt (pS473) antibody | BD Biosciences | Cat# 560378, RRID:AB_1645328 | As recommended by manufacturer |
| Antibody | BD Phosflow PerCP-Cy5.5 Mouse anti-ERK1/2 (pT202/pY204) antibody | BD Biosciences | Cat# 560115, RRID:AB_1645298 | As recommended by manufacturer |
| Antibody | BD Phosflow PE Mouse anti-PLK1 (pT210) antibody | BD Biosciences | Cat# 558445, RRID:AB_647227 | As recommended by manufacturer |

*Continued on next page*

*Continued*

| Reagent type (species) or resource | Designation | Source or reference | Identifiers | Additional information |
|---|---|---|---|---|
| Antibody | PE Goat Anti-Mouse Ig (Multiple Adsorption) | BD Biosciences | Cat# 550589, RRID:AB_393768 | (1:500) |
| Other | Biotinylated Sambucus Nigra Lectin (SNA) | Vector Laboratories | Cat# B-1305, RRID:AB_2336718 | |
| Other | Biotinylated Maackia Amurensis Lectin II (MAL II) | Vector Laboratories | Cat# B-1265, RRID:AB_2336569 | |
| Other | Streptavidin-FITC | eBioscience | Cat# 11-4317-87 | (1:500) |
| Other | PE Streptavidin | BD Biosciences | Cat# 554061, RRID:AB_10053328 | (1:500) |
| Commercial assay or kit | PE Annexin V | BD Pharmingen | Cat# 556422 | As recommended by manufacturer |
| Software | Imaris version 8.1 | Bitplane | RRID:SCR_007370 | |
| Recombinant DNA reagent | Mouse Siglec1 full length cDNA | Transomic technologies | Cat# BC141335 | Construction of expression plasmid |
| Transfected construct (mouse) | siRNA-1 to mouse Siglec1 | Bioneer | Cat# 20612–1 | RNA-GUC UUC CUU UCG AGA CUC A = tt(1-AS) |
| | | | | RNA-UGA GUC UCG AAA GGA AGA C = tt(1-AA) |
| Transfected construct (mouse) | siRNA-2 to mouse Siglec1 | Bioneer | Cat# 20612–2 | RNA-CUC CAA CCA ACU UCA CGA U = tt(2-AS) |
| | | | | RNA-AUC GUG AAG UUG GUU GGA G = tt(2-AA) |
| Transfected construct (mouse) | Negative control siRNA | Bioneer | Cat# SN-1003 | |
| Transfected construct (mouse) | pRGEN-CjCas9-CMV | Toolgen | TGEN_CjS1 | |
| Transfected construct (mouse) | pRGEN-U6-mSt3gal3-CjRG1 | Toolgen | TGEN_CjS1 | RGEN sequence: CAGTAAGTGTAGC TTCCAGGCAGAATAATA C |
| Sequence based reagent | mSt3gal3 For | This paper | PCR primers | TCACTATGCGGAGGAAGA CTGCTTAATATC |
| Sequence based reagent | mSt3gal3 Rev | This paper | PCR primers | ATGCAGATTTCAAGGGT TGGGGGAAG |

## Cell culture

Mouse melanoma cell line B16F10 (ATCC) and its derivatives, B16F10-GFP and *St3gal3* knockout B16F10-GFP cell lines, were cultured in DMEM supplemented with 10% FBS and 2 mM l-glutamine. J774A.1 (ATCC) mouse macrophage cells were cultured in DMEM supplemented with 10% FBS and 2 mM l-glutamine. 4T1 (ATCC) mouse breast cancer cells were cultured in RPMI-1640 medium supplemented with 10% FBS. HEK293T (ATCC) cells were cultured in DMEM medium supplemented with 10% FBS and 2 mM l-glutamine. Melan-A, a spontaneously immortalized melanocyte cell line, was kindly provided by Prof. Eun-Ju Chang, Asan Medical Center, Seoul, Korea. Melan-A cells were cultured in RPMI-1640 medium supplemented with 10% FBS, 200 nM phorbol 12-myristate 13-acetate (PMA), 50 μg/mL streptomycin, and 50 U/mL penicillin. All cells tested negative for mycoplasma.

## Generation of mouse ST3 beta-galactoside alpha-2,3-sialyltransferase 3 (*St3gal3*) knockout cell line

pRGEN-CjCas9-CMV and pRGEN-U6-mSt3gal3-CjRG1 plasmids were purchased from Toolgen (Seoul, South Korea) with the following target sequence of RGEN: CAGTAAGTGTAGCTTCCAGG-CAG<u>AATAATAC</u> (underlined are PAM sequence-NNNNRYAC- not included in gRNA but recognized by Cas9 protein). B16F10-GFP cells were transfected with the abovementioned plasmids (ratio 1:1) using Lipofectamine 3000. Forty-eight hours after transfection, cells were single-cell sorted on a FACSAria cell sorter (BD Biosciences, US) and cultured in 96-well plates to isolate genomic DNA and PCR-amplify target regions. The following primers were used for amplifying target sites by Sanger DNA sequencing to select the KO cell line: forward primer 5′-TCACTATGCGGAGGAAGACTGC TTAATATC-3′, reverse primer 5′-ATGCAGATTTCAAGGGTTGGGGGAAG-3′. Forward primer was used for the DNA sequencing.

## Animal studies

C57/BL6 and BALB/c mice were purchased from OrientBio (Gapyeong, Korea) and were between 6 and 8 weeks of age at use. Mice were housed in specific pathogenfree conditions at the National Cancer Centre, Goyang. All experiments using animals were performed in accordance with protocols (NCC-17–303C) approved by the Institutional Animal Care and Use Committee of the National Cancer Centre and in accordance with the Guide for the Care and Use of Laboratory Animals. To study early LN metastatic cells, we implanted $2 \times 10^5$ B16F10-eGFP or 4T1-eGFP subcutaneously into one hind limb footpad of each anesthetized mouse (Zoletil 40 mg/kg, Rompun 5 mg/kg). Animals were killed at different time points (starting from day 10) to collect popliteal LNs. For LN metastasis assays, anesthetized mice were subcutaneously implanted with $2 \times 10^5$ B16F10-GFP or *St3gal3* knockout B16F10-GFP cells in one hind limb footpad. Mice were killed at the end of the metastasis protocol, that is., 21 days for the LN assessment. For total metastatic burden in LNs, LNs were processed as described under tissue dissociation for FACS staining sections.

Primary tumor volumes were calculated using the following formula- volume (mm$^3$) = (L $\times$ W$^2$)/2, where L (length) is the larger of two perpendicular axes and W (width) is the smaller of two perpendicular tumor axes.

To confirm Siglec1 expression on the SCS macrophages, LN macrophages were depleted by footpad injection of 30 µL clodronate liposome. PBS liposomes were injected as control liposomes. Both liposomes were purchased from http://clodronateliposomes.org.

## Cell adhesion assay

Full-length mouse *Siglec1* cDNA (Transomic Technologies) was cloned into mammalian expression plasmid pd18. HEK293T cells were cultured in chamber slides to 70–80% confluency in DMEM supplemented with 10% FBS. At this point cells were transfected with empty vector (pd18) or vector expressing mouse Siglec1 (pd18-mSiglec1) using Lipofectamine 3000 (Invitrogen). After 6 hr transfection, medium was changed, and cells were cultured for three days before use in cell adhesion assays. On day 3, single cell suspensions of GFP-expressing cancer cells were prepared using enzyme-free PBS-based cell dissociation buffer (Gibco) and passing cells through 40 µm nylon filters (Falcon). Cells were counted and $1 \times 10^5$ cells in 0.5 ml DMEM supplemented with 10% FBS were added per chamber. Cells were incubated for 5 min at 37°C, and then the medium was removed with non-attached cells and wells were further washed with complete medium three times with gentle force with a P1000 pipette. Cells were fixed by adding 2% paraformaldehyde directly into wells and incubating for 10 min at 37°C. Cells were stained as described in the immunostaining section. After image acquisition, adherent cells were counted using the Spots tool in Imaris (version 8.1; Bitplane).

For extracellular matrix cell adhesion assays, wells in 96-well plates were coated with 5 µg/ml fibronectin overnight at 4°C. The next day, wells were washed with PBS and then with DMEM. Single cell suspensions were prepared using enzyme-free cell dissociation buffer and cells were passed through 40 µm nylon filters. A total of $5 \times 10^3$ cells per well were added in a volume of 100 µl DMEM and incubated for 30 min at 37°C. After incubation, wells were washed with DMEM to remove unbound cells and 100 µl DMEM supplemented with 10% FBS was added to each well and incubated for 2 hr to allow cells to recover. Next, 20 µl CellTiter 96 AQueous One Solution

(Promega) was added to each well and incubated for 1 hr at 37°C. After incubation, absorbance was measured at 490 nm using a 96-well plate reader. Fibronectin-coated wells with 100 µl DMEM supplemented with 10% FBS and without added cells were used to deduct the background.

## Immunohistochemical staining

LNs were fixed with 4% paraformaldehyde and processed for cryosectioning. OCT-embedded tissues were sectioned by cryostat (Leica) in thin (20 µm) or thick (100 µm) sections. Thin sections were used to locate and visualize pioneer metastasis cells. All sections were inspected under the Zeiss Axio Imager Z1 microscope to locate sections with GFP-positive cells. For antibody staining, LN sections were permeabilized and blocked in 5% goat serum in PBST (0.3% Triton X-100 in PBS). Antibodies used for immunohistochemical staining are listed in the Supplementary Information. For staining in in vitro cell adhesion assays, cells in chamber slides were fixed with 2% paraformaldehyde and stained using a similar protocol as the LN staining. Sections were mounted in fluorescent mounting medium (Dako) and images were acquired on a LSM 780 and LSM 880 Laser Scanning Confocal Microscope (Carl Zeiss SAS, Jena, Germany). All images were processed on Imaris (version 8.1; Bitplane) and shown as maximum intensity projections. *Figure 3D* reconstruction was performed on Imaris using the surface tool.

## Co-culture experiments

HEK293T cells were grown in DMEM with 10% FBS, and 70–80% confluent cells were used for transfection with mammalian expression plasmids pd18 (empty plasmid; mock) or pd18-mSiglec1 using Lipofectamine 3000 (Invitrogen). Medium was changed to CDM293 (Invitrogen) supplemented with L-glutamine and cells were further cultured for 3 days to ensure maximum mSiglec1 expression on the cell surface. On day 3, single-cell preparations of B16-GFP cells were prepared by dissociating cells with enzyme-free PBS-based cell dissociation buffer (Gibco) and passing cells through 40 µm nylon filters (Falcon). Both HEK293T cells and cancer cells were counted and mixed at a 4:1 ratio in DMEM supplemented with 5% FBS. Cells were cultured in low binding plates (Corning) for 18 hr at 37°C in 5% $CO_2$. At the end of the experiments, cells either underwent cell sorting to assess gene expression or were fixed in Cytofix buffer for Ki67 or Phosflow staining. For apoptosis quantification PE Annexin V (BD biosciences) was used according to manufacturer's instructions.

## siRNA transfection into J774A.1 cells and co-culture experiment

Pre-designed negative control and mouse *Siglec1* siRNAs were obtained from Bioneer (Bioneer, Daejeon, Korea). J774A.1 cells were reverse-transfected with siRNA using Lipofectamine RNAiMAX transfection reagent (Invitrogen) in Opti-MEM medium. Medium was changed after 6 hr post-transfection to DMEM supplemented with 10% FBS. Siglec1 expression was assessed on day 3 and day four post-transfection using Alexa647-conjugated anti-mouse Siglec1 antibody. Co-culture experiments were performed as described in the previous section.

## Ki67 and phosflow analysis

For Ki67 and Phosflow (pAKT, pERK, and pPLK1) analysis in co-culture experiments, cells were counted and fixed by adding Cytofix buffer directly into the cell culture, and cells were incubated for 10 min at 37°C. After washing with PBS, cells were permeabilized in 0.1% Triton X-100 in PBS for 5 min at room temperature. After washing the cells in staining buffer, the cells were incubated with mouse FcR blocking reagent (Miltenyi Biotec). Then samples were incubated with Ki67 or Phosflow antibody in staining buffer. Following incubation, cells were washed and analyzed by FACSverse (BD); data were assessed using gates for GFP$^+$ cells. Fold changes were calculated by dividing the mean fluorescence intensity of experimental samples by the mean fluorescence intensity of control samples.

## Tissue dissociation for FACS staining

LNs and primary tumors were mechanically dissociated with forceps and scalpels and placed in DMEM containing 2 mg/ml collagenase (Sigma) for 20 min at 37°C (for LN metastatic burden evaluation, collagenase incubation time was increased to 40 min). The suspensions were incubated in 0.1 mg/ml DNase I (Roche) for 10 mins. Tissue suspensions were resuspended with P1000, filtered

through 40 µm nylon filters (Falcon), and aliquoted for total cell counting to adjust equal cell numbers for antibody staining. For antibody staining and FACS analysis, cells were fixed in 1 × Cytofix fixation buffer (BD) for 10 min at 37°C. After washing with PBS, cells were permeabilized in 0.1% Triton X-100 in PBS for 5 min at room temperature. Cells were washed and resuspended in staining buffer (BD) and then blocked with mouse FcR blocking reagent. Then samples were incubated with FACS antibodies and processed as described in the previous section.

### Biotinylated lectin staining for FACS

For lectin staining, $1 \times 10^5$ cells were resuspended in lectin staining buffer (10 mM HEPES, pH 7.5, 0.15M NaCl, 0.09% sodium azide) stained with 1 µg/ml lectin in a 100 µl volume at 4°C for 30 min followed by incubation with detection reagent streptavidin-PE for 20 min at 4°C.

### Sialidase treatment

A total of $5 \times 10^5$ B16F10 cells were treated with 0.5 U/ml sialidase (from *Clostridium perfringens*; Roche Applied Science) in serum-free RPMI medium for 1 hr at 37°C in 5% $CO_2$ and 95% relative humidity. Control cells were incubated in similar conditions but without the enzyme. After 1 hr, cells were washed in lectin staining buffer and stained with lectin or mouse Siglec1(ECD)-mFc.

### Transwell migration assay

Wild type and *St3gal3* knockout B16F10-GFP cells were serum-starved overnight in DMEM medium. The next day, a single cell suspension was prepared using enzyme-free cell dissociation buffer and $5 \times 10^4$ cells were added in 100 µl to the upper chambers of 24-well transwell permeable supports with 8 µm pores (Corning). The lower chambers were filled with 650 µl DMEM medium supplemented with 10% FBS as a chemoattractant. Control chambers were filled with DMEM without FBS. After 8 hr of incubation, the upper chambers were washed with PBS and non-migrated cells from the upper side of transwell chambers were removed gently using cotton swabs. Then cells on the lower side of the membranes were fixed with methanol for 10 min. Cells were stained with crystal violet solution for 10 min and thoroughly rinsed with water. After drying the membranes, images were captured on a Zeiss Axio Imager M1 (Carl Zeiss) microscope using a 10 × objective. The number of total migrated cells was counted using the following formula: (average number on cells per image/image area) × 0.33 cm$^2$. Next, the percentage of migrated cells was counted using the following formula: (number of migrated cells/50,000) × 100.

### Metastatic burden of LNs

To calculate the total number of GFP$^+$ metastatic cells in LNs, the total number of LN cells was counted after tissue digestion. The number of metastatic cells was determined by gating around GFP$^+$ cells and the following formula was used to calculate the final number of metastatic cells in LNs: (number of cells in GFP$^+$ gate/total number of FACS analyzed cells) × total number of LN cells.

### Transcriptome analysis

After 18 hr of co-culture of GFP-expressing cancer cells with Siglec1-expressing or mock HEK293T cells in low binding plates, B16-GFP cells were sorted using a BD FACS Aria II or a BD FACS Aria SORP based on GFP expression. Total RNA from samples was isolated using a RNeasy Plus Mini kit (Qiagen). Library preparation for RNA sequencing and data analysis was performed by Macrogen (Macrogen Inc, Korea). Briefly, a TruSeq RNA library (Illumine) was constructed according to the manufacturer's instructions and RNA-seq was performed on a HiSeq4000 (Illumina). Processed trimmed reads were mapped to known reference genomes through the TopHat program (version 2.0.13) and then aligned to UCSC mm10 using Bowtie aligner (bowtie2 v2.2.3), which allows splice junction processing. After read mapping, transcript assembly work was performed through the cufflinks program (version 2.2.1). As a result, the expression profile values for each sample were obtained for each transcript, and the FPKM (Fragment per Kilobase of Transcript per Million mapped reads) values were compiled based on the gene transcripts. *Siglec1* and *Dhfr* were not included in the differential expressed gene counting and pathway analysis due to their plasmid origin. KEGG pathway analysis of unregulated genes was performed using the database for annotation, visualization and integrated discovery (DAVID) v6.8 (**Huang et al., 2009a**; **Huang et al., 2009b**). GO

enrichment analysis of unregulated genes was performed using the Gene Ontology (GO) knowledgebase (*Mi et al., 2017*).

### Statistical analysis

All data are expressed as mean ±s.d. and data points are shown as scatter plots. Significance between two groups was analyzed with two-tailed, unpaired t-tests using Graphpad Prism software. Exact *P*-values are shown in the graphs.

## Acknowledgements

We thank Prof. Eun-Ju Chang (Asan Medical Center, Seoul, Korea) for providing mouse Melan-A cell line, Tae-Sik Kim, Flow Cytometry Unit and Mi Ae Kim, Microcopy Imaging Unit, National Cancer Center for technical support. We also thank Hye Young Seo for critical reading of manuscript and comments.

## Additional information

### Funding

| Funder | Grant reference number | Author |
| --- | --- | --- |
| National Cancer Center | 1810102 | Beom K. Choi |
| National Cancer Center | 1910050 | Beom K. Choi |

The funders had no role in study design, data collection and interpretation, or the decision to submit the work for publication.

### Author contributions

Rohit Singh, Conceptualization, Formal analysis, Validation, Investigation, Visualization, Methodology, Writing—original draft, Writing—review and editing; Beom K Choi, Resources, Formal analysis, Supervision, Funding acquisition, Validation, Investigation, Methodology, Writing—original draft, Writing—review and editing

### Author ORCIDs

Rohit Singh https://orcid.org/0000-0003-3923-144X

### Ethics

Animal experimentation: All experiments using animals were performed in accordance with protocols (protocol number NCC-17-303C) approved by the Institutional Animal Care and Use Committee of National Cancer Centre and were performed in accordance with the Guide for the Care and Use of Laboratory Animals.

### Decision letter and Author response

Decision letter https://doi.org/10.7554/eLife.48916.sa1
Author response https://doi.org/10.7554/eLife.48916.sa2

## Additional files

### Supplementary files

• Transparent reporting form

### Data availability

All data are available in the main paper or the supplementary materials. RNA-seq data have been deposited in NCBI Gene Expression Omnibus (GEO) under accession number GSE109077.

The following dataset was generated:

| Author(s) | Year | Dataset title | Dataset URL | Database and Identifier |
|-----------|------|---------------|-------------|--------------------------|
| Singh R, Choi BK | 2019 | Analysis of transcriptome of mouse melanoma cells co-cultured with HEK293T cells expressing mouse Siglec1 | https://www.ncbi.nlm.nih.gov/geo/query/acc.cgi?acc=GSE109077 | NCBI Gene Expression Omnibus, GSE109077 |

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
