## [Decision Letter]

**Acceptance summary:**

In this study, the authors attempt to understand the mechanisms underlying lymph node metastasis, a common site for metastasis for disease such as melanoma. They demonstrate that GFP tagged pioneer metastatic cells interact with Siglec1+ subcapsular sinus macrophages, and that this interaction promotes their proliferation. Importantly, knockout of the St3gal3 sialyltransferase compromised the metastatic efficiency of tumor cells by reducing α-2,3-linked sialylation. These results suggest a mechanism by which tumor cells can coopt a hypersialylated mechanism to promote distant colonization, which may be a paradigm for how other tumors do this as well.

**Decision letter after peer review:**

Thank you for submitting your article "Siglec1-expressing subcapsular sinus macrophages provide soil for melanoma lymph node metastasis" for consideration by *eLife*. Your article has been reviewed by two peer reviewers, and the evaluation has been overseen by a Reviewing Editor and Päivi Ojala as the Senior Editor. The reviewers have opted to remain anonymous.

The reviewers have discussed the reviews with one another and the Reviewing Editor has drafted this decision to help you prepare a revised submission.

In this manuscript, Singh and Choi present data that macrophages in the lymph nodes express Siglec1/CD169, which interacts with sialylated proteins on the surface of B16F10 cells. These data may provide further information as to the mechanism by which melanoma cells colonize lymph nodes, an important initial site of metastasis in this disease. Overall, the data are of interest and will open the way for further investigations. There are several areas the reviewers felt would heighten the impact/generalizability of the data and solidify its importance for the field. Specifically:

Major requests:

1) Signaling downstream of Siglec1

Confocal microscopy shows close association between Siglec1+ macrophages and tumour cells. However, this does not itself prove that signalling mediated by Siglec1/sialylated protein interaction is occurring directly between these cell types. The in vitro experiments showing the responses of B16F10 cells to Siglec1 signalling are weakened by the fact that they are conducted using HEK-293 cells to express Siglec1. Mouse macrophages and human HEK-293 cells would express very different ranges of secreted and cell-surface molecules that could modulate the outcome of Siglec1 signalling. It would be better to use SCS macrophages if possible, or otherwise a closely matched mouse macrophage cell line, for these in vitro experiments. Siglec1 could then be either overexpressed or knocked down. It may only be necessary to do a few key in vitro experiments with SCS macrophages to validate the general findings established using HEK-293 cells.

Is treatment with recombinant siglec-1 sufficient to cause downstream PI3K/Akt or Erk signaling and proliferation? Or do these signaling pathways require cell surface siglec-1?

2) The effects of volasertib

Is it surprising that blocking the cell cycle with volasertib inhibited growth of metastatic cancer cells in LNs? It is unclear how this relates to the proposed siglec-1 mechanism. Further, the examination of primary growth is not indicative of LN specificity of volasertib as the drug was only delivered locally. If volasertib is given systemically, what are the effects on primary tumor growth? Staining of the primary tumour for cleaved Caspase-3/Ki67 would help rule in or out any effects on metastasis secondary to effects on the primary tumour for these experiments.

3) Cell-intrinsic properties of Siglec1

Several other possible explanations for impaired metastatic colony formation of St3gal3 KO tumour cells need to be investigated. Does knockout of St3gal3 affect inherent adhesive, migratory or invasive properties of B16F10 cells? Is there Siglec1 expressed in the primary tumour (e.g. in lymphatics) that might affect the initial rate of metastatic tumour cells escaping from the primary tumour? Comparison of% Ki67+ between St3gal3 WT and KO tumour cells in the SCS would also be informative.

4) Generalizability of the findings

Are the results of the CD169/SCS macrophage mechanism specific to B16F10 cells or is the mechanism of LN colonization found in other cell lines?

5) Specificity of Siglec1 on macrophages

While the abundant expression of Siglec1 on SCS macrophages is well-established, Iftakhar et al. (2016, PNAS) and other studies have indicated some (albeit lower) expression of Siglec1 in SCS floor lymphatic endothelial cells (LECs). An appropriate macrophage-specific marker (e.g. CD11b) should be used in combination with LYVE1 to confirm a) the identity of the Siglec1-expressing cells interacting with tumour cells, and b) what proportion of the Siglec1 staining in the LN belongs to macrophages.

6) How the Siglec1 interaction promotes metastasis

"As we observed a shift towards increased cell cycle activity and proliferation in Siglec1-interacting cancer cells, and abolition of this interaction in vivo revealed the vulnerability of early metastasis colonization process." What in vivo abolition of siglec-1 interaction is this statement referring to? The only interventions at this point in the manuscript seem to be performed in vitro, not in vivo making this statement confusing. To what are the authors referring?

Quantification of several key results:

1) Figure 2: There is no quantification of cleaved caspase 3 positive cancer cells to support the authors' conclusion. These data should be quantified. Further, why was cleaved caspase-3 not measured in the co-culture experiments as was done for Ki67? It seems the lack of binding of cancer cells would lead to apoptosis, making cleaved caspase-3 a relevant measure.

2) How reliable is visual inspection for metastasis in melanoma? The authors should analyze tissue sections to identify microscopic metastasis and verify that metastatic nodes visually identified did indeed contain tumor cells and those judged to be negative were cancer-free. It is known that determining the incidence of metastasis based only on the presence of melanin is unreliable. A microscopic examination is needed.

3) Quantitation of% Ki67+ GFP cells as single cells vs those in colonies would be helpful to support the rest of Figure 2 and the claims that tumour cells begin proliferating after contact with Siglec1+ macrophages (see also point 3).

[Editors' note: further revisions were requested prior to acceptance, as described below.]

Thank you for submitting your article "Siglec1-expressing subcapsular sinus macrophages provide soil for melanoma lymph node metastasis" for consideration by *eLife*. The reviewers have discussed the reviews with one another and the Reviewing Editor has drafted this decision to help you prepare a revised submission.

In general, the reviewers felt that you addressed most of their previous concerns, but that a few issues remain. One significant question was how generalizeable the findings were, and there is concern that the 4T1 cell experiments did not fully address this. We do not feel you have to necessarily do additional validation experiments, but if you have additional data I would suggest you add it to the manuscript or add text to the document addressing this point. Other specific points, most of which can be addressed by clarifying the Figures or text, are listed below:

Reviewer #1:

Origins of 4T1 breast cancer cell line not in Key Resources Table or in Materials and methods. Also, it appears that the 4T1 breast cancer cells were grown subcutaneously and not in the mammary fat pad. Why was this done? Is the Siglec mediated binding only a result of tumors that are grown in the skin? Were 4T1 only used in one experiment, with no further verification of the proposed mechanism? It seems none of the co-culture experiments with Siglec overexpressing human macrophages or Siglec downregulated murine macrophages were performed with another cell line like 4T1 to determine whether the signaling is specific only to B16 cells. The presented experiments with 4T1 do not address the reviewer's questions about how generalizable the hypersialation and Siglec mechanism would be to other cell lines.

In Figure 1, examples of cancer cells sitting on SCS macrophages are shown. Are there examples where cancer cells are directly on LECs? The staining seems to show that the SCS macrophages form an almost complete barrier, suggesting that interaction of cancer cells on the SCS layer is not an active process but just result of the density of SCS macrophages.

Figure 1—figure supplement 3 is missing a LEC stain identify the SCS floor?

Figure 1—figure supplement 4 A and B. It appears that the graph key is incorrect. As it is currently shown, Melan-A has significantly more sialyation than B16F10, which contradicts the graphs in C and D as well as what is written in the text.

Subsection “Siglec1-interacting cancer cells show higher proliferation”: "As our data confirmed adhesive cell-to-cell contact between SCS macrophages and pioneer metastatic cells…" The data do not confirm adhesive cell-cell contact in vivo. The data show contact and proximity, but not adhesion. The in vitro assay did not use only pioneering metastatic cells, but a cell line from culture. This statement needs to be revised. Further, the author use the term pioneering metastatic cells frequently, but the term should only be used for the earliest arriving cells to the lymph node in vivo, and not to cells in a cell culture dish.

Figure 2: Why are the SSC-A values so different for panels F and H? They should be from similar experimental conditions. Panels J-M need strong positive controls for pERK and pAKT. There is not much of a shift in pERK and pAKT and the biological relevance of these data are questionable at best.

Figure 2—figure supplement 2. Why are the SSC-A values so different for panels E and G? They should be from similar experimental conditions. In Panels I and K, it is very challenging to see how the pERK and pAKT are actually different in any meaningful way. Are these findings robust? Can a strong positive control be used for pERK and pAKT to show what a true positive should look like? Otherwise these data are not very strong.

Reviewer #2:

Overall the authors have addressed most of the issues that were raised.

We had questioned (in point #5) the evidence in the data for the point that the disruption of the macrophage-lymphatic SCS lining by the growing metastasis is part of the colonisation process. This point was not directly addressed in the Response letter, but some additional text has been added to tone down their claims. But directly addressing this concern in the response would be more informative.

Major requests:

1) Signaling downstream of Siglec1

The authors have addressed this concern by performing co-culture experiments of B16F10 cells with the mouse macrophage line J774A.1 to model SCS macrophages. J774A.1 endogenously expresses Siglec1, and siRNA-mediated knockdown of the endogenous protein resulted in increased apoptosis and decreased proliferation of co-cultured B16F10 cells. This complements the prior results obtained using overexpression of Siglec1 in HEK293 cells. The authors should ensure that for figures concerning the siRNA data, the x-axis is labelled as siRNA (Figure 4G, Figure 2—figure supplement 2 F, H, L) to avoid confusion with the HEK293 overexpression data.

2) The effects of volasertib

The removal of this experiment does help streamline the manuscript and retain focus on Siglec1-mediated downstream signaling.

3) Cell-intrinsic properties of Siglec1

The experiments performed confirm that altering st3gal3 expression does not have any intrinsic effects on B16F10 cell behaviour, and validated that abrogating siglec1-mediated signaling impairs proliferation and survival of metastasising cells in the SCS.

The lack of staining for Siglec in tumour lymphatics shown in the response letter only is somewhat superficial and does not appear to be mentioned in the new document text.

4) Specificity of Siglec1 on macrophages

It would have been useful to see direct co-staining for Siglec1 and a macrophage marker. There does appear to be faint Siglec staining on LYVE1+ LECs in the medullary sinuses of macrophage-depleted LNs in Figure 1—figure supplement 2B; however the new experiments performed do indicate that the vast majority of Siglec1 in LNs is expressed on sinusoidal macrophages. The results text should describe the macrophage depletion experiment – currently this appears not to be specifically mentioned.

---

## [Author Response]

Major requests:1) Signaling downstream of Siglec1

*Confocal microscopy shows close association between Siglec1+ macrophages and tumour cells. However, this does not itself prove that signalling mediated by Siglec1/sialylated protein interaction is occurring directly between these cell types. The* in vitro *experiments showing the responses of B16F10 cells to Siglec1 signalling are weakened by the fact that they are conducted using HEK-293 cells to express Siglec1. Mouse macrophages and human HEK-293 cells would express very different ranges of secreted and cell-surface molecules that could modulate the outcome of Siglec1 signalling. It would be better to use SCS macrophages if possible, or otherwise a closely matched mouse macrophage cell line, for these* in vitro *experiments. Siglec1 could then be either overexpressed or knocked down. It may only be necessary to do a few key* in vitro *experiments with SCS macrophages to validate the general findings established using HEK-293 cells.*

We thank the reviewers for pointing this out. Given the difficulty in isolating sufficient numbers of SCS macrophages, as well as there being no standard protocol for their culture, we performed experiments with the mouse macrophage cell line J774A.1. J774A.1 cells express Siglec1 and thus we performed siRNA-mediated knockdown of *Siglec1* in this line for co-culture experiments. The results are provided in Figure 2—figure supplement 2 and in Figure 4E–G. Consistent with our in vitro findings in HEK293 cells, we observed similar results using a comparable line when we used *Siglec1*-knocked down J774A.1 cells for co-culture experiments.

Is treatment with recombinant siglec-1 sufficient to cause downstream PI3K/Akt or Erk signaling and proliferation? Or do these signaling pathways require cell surface siglec-1?

During the initial experiments, we used recombinant mouse Siglec1. However, the magnitude of change was very low and was mostly non-significant (data not shown). Nonetheless, co-culture experiments better reflect the setting of metastasis in which cell-to-cell interactions take place. Therefore, we persisted with this set-up rather than further exploring the use purified protein in our experiments.

2) The effects of volasertibIs it surprising that blocking the cell cycle with volasertib inhibited growth of metastatic cancer cells in LNs? It is unclear how this relates to the proposed siglec-1 mechanism. Further, the examination of primary growth is not indicative of LN specificity of volasertib as the drug was only delivered locally. If volasertib is given systemically, what are the effects on primary tumor growth? Staining of the primary tumour for cleaved Caspase-3/Ki67 would help rule in or out any effects on metastasis secondary to effects on the primary tumour for these experiments.

As the reviewers point out, metastasis inhibition by volasertib was not a surprising observation, and, this particular set of experiments was not directly linked to the Siglec1 mechanism. Therefore, to ensure clarity and maintain the focus of the revised manuscript on Siglec1, we have removed the volasertib experiment from Figure 4. Rather, we have focused this section on the high mitotic activity measured by PLK1 phosphorylation in early metastatic cells, and demonstrated that Siglec1-interacting cancer cells showed higher mitotic commitment compared with cells that did not interact with Siglec1. These data are shown in Figure 4.

3) Cell-intrinsic properties of Siglec1Several other possible explanations for impaired metastatic colony formation of St3gal3 KO tumour cells need to be investigated. Does knockout of St3gal3 affect inherent adhesive, migratory or invasive properties of B16F10 cells? Is there Siglec1 expressed in the primary tumour (e.g. in lymphatics) that might affect the initial rate of metastatic tumour cells escaping from the primary tumour? Comparison of% Ki67+ between St3gal3 WT and KO tumour cells in the SCS would also be informative.

Thank you for your comments. We performed additional experiments to assess the adhesion and transwell migration of WT and *St3gal3* KO tumor cells. These data have been provided in Figure 5—figure supplement 1H–J. We did not find any significant difference in the in vitro adhesion and migration ability of St3gal3 WT and KO tumor cells.

We performed immunohistochemistry and confocal imaging to explore Siglec1 expression in tumor lymphatic endothelial cells but none was evident. This discounted the possibility of Siglec1 expression on tumor lymphatic cells and its affect on the initial rate of metastatic cells escaping from the primary tumor. See Figure 5—figure supplement 2.

We thank the reviewers for suggesting a comparison of positive Ki67 expression in *St3gal3* WT and KO tumor cells in the SCS. We have provided a new set of data showing percentage of Ki67 positivity in the early stages of metastatic colonization in Figure 5G, H. This reduced proliferation and increased apoptosis shown in Figure 5G–J in *St3gal3* KO early metastatic cells/foci supports our observation that Siglec1-expressing SCS macrophages-pioneer metastatic cell interaction is required for efficient LN metastatic colonization.

4) Generalizability of the findingsAre the results of the CD169/SCS macrophage mechanism specific to B16F10 cells or is the mechanism of LN colonization found in other cell lines?

To establish if LN metastasizing cancer cells other than melanoma also use SCS macrophages, we used GFP-expressing mouse breast cancer cells. Due to low GFP intensity, we detected the 4T1 cells using an anti-GFP antibody. We confirmed that breast cancer cells were also anchored by Siglec1-expressing SCS macrophages. These new data are provided in Figure 1—figure supplement 3. In light of our findings in melanoma as well as LN metastasis with early metastatic events observed by Das et al. and Jeong et al. (ref. 11, 12 in manuscript) using squamous cell carcinoma, breast cancer, and melanoma, it is possible that other LN-metastasizing tumors also use a similar strategy and LN SCS macrophages provide “soil” to other tumor types.

5) Specificity of Siglec1 on macrophagesWhile the abundant expression of Siglec1 on SCS macrophages is well-established, Iftakhar et al. (2016, PNAS) and other studies have indicated some (albeit lower) expression of Siglec1 in SCS floor lymphatic endothelial cells (LECs). An appropriate macrophage-specific marker (e.g. CD11b) should be used in combination with LYVE1 to confirm a) the identity of the Siglec1-expressing cells interacting with tumour cells, and b) what proportion of the Siglec1 staining in the LN belongs to macrophages.

As suggested by the reviewers, we have provided additional data using CD11b and Lyve1 to confirm that the interaction in the LN SCS is indeed between pioneer metastatic cells and CD11b^+^ SCS macrophages. Data have been provided in Figure 1—figure supplement 1C.

To address query (b) with regards to what proportion of Siglec1 staining belongs to macrophages, we depleted LN macrophages using clodronate liposomes with PBS liposomes as a control. Siglec1 and Lyve1 staining of control and macrophage-depleted LNs showed that Siglec1 is primarily expressed on SCS and medullary sinus macrophages. Of note, in macrophage-depleted LNs we could not detect Siglec1, which could account for the interaction that we observed between metastatic cells and SCS macrophages by Siglec1 staining. Data have been provided as Figure 1—figure supplement 2.

6) How the Siglec1 interaction promotes metastasis

*"As we observed a shift towards increased cell cycle activity and proliferation in Siglec1-interacting cancer cells, and abolition of this interaction* in vivo *revealed the vulnerability of early metastasis colonization process." What* in vivo *abolition of siglec-1 interaction is this statement referring to? The only interventions at this point in the manuscript seem to be performed* in vitro*, not* in vivo *making this statement confusing. To what are the authors referring?*

We thank reviewers to bringing this error to our attention. Originally, Figure 3 was placed after the *St3gal3* KO B16-GFP cell line experiments and this statement was made in that context. During the final drafting of the manuscript this error went unnoticed by us. We have now fixed the error in the revised manuscript.

Quantification of several key results:1) Figure 2: There is no quantification of cleaved caspase 3 positive cancer cells to support the authors' conclusion. These data should be quantified. Further, why was cleaved caspase-3 not measured in the co-culture experiments as was done for Ki67? It seems the lack of binding of cancer cells would lead to apoptosis, making cleaved caspase-3 a relevant measure.

We have provided the quantification of cleaved caspase-3-positive cells in the LN SCS in Figure 2B. Additionally, we have provided quantification of Ki67-positive cells in the LN SCS in Figure 2D.

Additionally, we thank the reviewers for raising this logical point regarding apoptosis measurement in co-culture experiments. We performed the additional experiments to measure apoptosis using Annexin-V-PE. These data have been added to Figure 2H, I (with HEK cells) and Figure 2—figure supplement 2G, H (with J774A.1 cells).

2) How reliable is visual inspection for metastasis in melanoma? The authors should analyze tissue sections to identify microscopic metastasis and verify that metastatic nodes visually identified did indeed contain tumor cells and those judged to be negative were cancer-free. It is known that determining the incidence of metastasis based only on the presence of melanin is unreliable. A microscopic examination is needed.

As mentioned in the response to major request #2, we have removed this dataset to ensure clarity and maintain the focus of the manuscript on Siglec1-mediated events.

3) Quantitation of% Ki67+ GFP cells as single cells vs those in colonies would be helpful to support the rest of Figure 2 and the claims that tumour cells begin proliferating after contact with Siglec1+ macrophages (see also point 3).

We have provided the quantification in Figure 2D.

[Editors' note: further revisions were requested prior to acceptance, as described below.]

Reviewer #1:Origins of 4T1 breast cancer cell line not in Key Resources Table or in Materials and methods. Also, it appears that the 4T1 breast cancer cells were grown subcutaneously and not in the mammary fat pad. Why was this done? Is the Siglec mediated binding only a result of tumors that are grown in the skin? Were 4T1 only used in one experiment, with no further verification of the proposed mechanism? It seems none of the co-culture experiments with Siglec overexpressing human macrophages or Siglec downregulated murine macrophages were performed with another cell line like 4T1 to determine whether the signaling is specific only to B16 cells. The presented experiments with 4T1 do not address the reviewer's questions about how generalizable the hypersialation and Siglec mechanism would be to other cell lines.

We have added the origin of the 4T1 breast cancer cell line to the Key Resources Table and thank the reviewer for pointing out this issue.

As the focus of our study was melanoma lymph node (LN) metastasis, we were able to determine the timeline of early metastatic cell arrival to popliteal LNs after several experiments. We had no such established protocol/timeline for the mammary fat pad injection experiments from which to determine the appropriate time point that pioneer metastatic cells may land in LNs. As subcutaneous injection of 4T1 cells is reported in the study of LN metastasis, we opted for this route and used our melanoma metastasis timeline as a guide [Hu et al., *Theranostics* 8(3), 3597-3610 (2018)]. We believe 4T1 metastatic cells originating from subcutaneous or mammary fat pad primary tumors colonize the LNs through lymphatic dissemination in a similar fashion, although the timeline might differ due to the different primary tumor growth rates. Of note, Brown et al. [Science 359, 1408-1411 (2018); reference 4 in the main manuscript] demonstrated that 4T1 mammary carcinoma cells even directly injected into popliteal LN afferent lymphatics by intralymphatic microinfusion show a normal lymph node colonization sequence.

Since we have examined the whole LN colonization process in the context of melanoma (which is reflected in the title of the manuscript), we performed detailed experiments with melanoma cell lines. In response to the reviewer’s query regarding the generalizability of the observations, we examined the initial event at which the metastatic cells landed in the LNs using GFP-expressing 4T1 mouse mammary carcinoma cells to determine whether breast cancer cells, which also frequently metastasize to LNs, also use SCS macrophages for anchorage. In the revised manuscript, we have strictly confined our statement to the similarity of the initial LN metastatic colonization step. We draw from the observations made by Das et al. and Jeong et al. (references 11 and 12 in manuscript, respectively) whereby used intravital microscopy and different tumor cells in mice to observe that LN metastatic cells stay and proliferate in LN SCS before invading the LN sinus, we speculated that breast cancer may also use a similar mechanism to melanoma. We have edited the manuscript text to reflect this inference. Further study is warranted to ascertain that other tumor types also use a similar mechanism.

In Figure 1, examples of cancer cells sitting on SCS macrophages are shown. Are there examples where cancer cells are directly on LECs? The staining seems to show that the SCS macrophages form an almost complete barrier, suggesting that interaction of cancer cells on the SCS layer is not an active process but just result of the density of SCS macrophages.

Thank you for your comment. Video 1 submitted with the manuscript shows pioneer metastatic cell in close proximity/sitting on the SCS lymphatic endothelial lining of SCS. In our multiple experiments, we observed metastatic cells at various distances from lymphatic endothelial cells that still had close interactions with SCS macrophages, even in low SCS macrophage-dense regions. Additionally, when we observe a SCS macrophage–lymphatic layer, e.g. in Figure 1—figure supplement 2, whereby the SCS macrophages did not form a complete barrier over the lymphatic lining of the SCS sinus in such a way as to leave no gaps for metastatic cancer cells to access the lymphatics lining. The in vivo observations of the SCS macrophage–pioneer metastatic cell interactions, combined with in vitro adhesion assays of Siglec1-overexpressing HEK293T and cancer cells, supports the active contact between Siglec1-expressing SCS macrophages and cancer cells. Furthermore, adhesion of cancer cells to Siglec1-overexpressing HEK293T cells was abolished after KO of *St3gal3* (which produces Siglec-bindig sialylation), further supporting the Siglec1-specific interaction with tumor cells.

Figure 1—figure supplement is missing a LEC stain identify the SCS floor?

The 4T1-GFP cell line used in our experiments showed very weak GFP signals after paraformaldehyde tissue fixation. Thus, to visualize the 4T1 cells, we had to amplify the GFP signals using rabbit anti-GFP antibody. Since the Lyve-1 antibody was also of rabbit origin, we were left with no option. We mention the use of the anti-GFP antibody in the figure legend of Figure 1—figure supplement 3.

Figure 1—figure supplement 4 A and B. It appears that the graph key is incorrect. As it is currently shown, Melan-A has significantly more sialyation than B16F10, which contradicts the graphs in C and D as well as what is written in the text.

We thank reviewer to pointing out our error and have corrected this oversight in the revised manuscript.

*Subsection “Siglec1-interacting cancer cells show higher proliferation”: "As our data confirmed adhesive cell-to-cell contact between SCS macrophages and pioneer metastatic cells…" The data do not confirm adhesive cell-cell contact* in vivo*. The data show contact and proximity, but not adhesion. The* in vitro *assay did not use only pioneering metastatic cells, but a cell line from culture. This statement needs to be revised. Further, the author use the term pioneering metastatic cells frequently, but the term should only be used for the earliest arriving cells to the lymph node* in vivo*, and not to cells in a cell culture dish.*

Thank you for your comments. We have modified the relevant sentence by deleting the word “adhesive”. We have also used the term “cell-to-cell contact” rather than “adhesive cell-to-cell contact” to describe the interaction between SCS macrophages and pioneer metastatic cells.

Furthermore, we thank the reviewers highlighting the use of the term “pioneer metastatic cell”. We have replaced it with “cancer cells” in two places and have kept it in places where we have inferred the effect in in vivo metastatic cells based on in vivo and/or in vitro data.

Figure 2: Why are the SSC-A values so different for panels F and H? They should be from similar experimental conditions. Panels J-M need strong positive controls for pERK and pAKT. There is not much of a shift in pERK and pAKT and the biological relevance of these data are questionable at best.Figure 2—figure supplement 2. Why are the SSC-A values so different for panels E and G? They should be from similar experimental conditions. In Panels I and K, it is very challenging to see how the pERK and pAKT are actually different in any meaningful way. Are these findings robust? Can a strong positive control be used for pERK and pAKT to show what a true positive should look like? Otherwise these data are not very strong.

Thank you for your comments. Although the data were from a similar co-culture experimental setting, the sample staining procedure and FACS analysis strategy was different. For the Ki67 analysis, we fixed and permeabilized the cells before staining, and for the FACS analysis, cells were gated on forward scatter (FSC) and side scatter (SSC) to eliminate debris. Whereas, for Annexin V staining, live cells were stained (without fixation and permeabilization) and no gating was applied. This was performed to include all live, apoptotic, and dead cells (application note: https://www.bdbiosciences.com/documents/BD_FACSVerse_Apoptosis_Detection_AppNote.pdf). As apoptotic cells show increased granularity, it was important to include high SSC-A cells in the analysis. Therefore, the SSC-A axis in the plots in Figure 2H and Figure 2—figure supplement 2G (Annexin V staining) shows more cells than that in the plots in Figure 2F and Figure 2—figure supplement 2E (Ki67 staining).

We would like to draw the reviewer’s attention to the fact that the Phosflow product data available on BD Biosciences’ website details a specific procedure, whereby cells are first serum-starved and then stimulated with strong AKT or ERK phosphorylation inducers to generate maximum changes in their phosphorylation status. In contrast in our experiments, at no point were the cells serum-starved, and there was a constant presence of FBS-derived growth factors in the culture media. This would not produce extreme changes in the phosphorylation status of AKT and ERK. Additionally, we needed to view our phosphorylation data in the context of the whole experimental setting in which we evaluated Ki67 (proliferation) and Annexin-V (apoptosis) in addition to Phosflow to understand the signaling mechanism.

Reviewer #2:Overall the authors have addressed most of the issues that were raised.We had questioned (in point #5) the evidence in the data for the point that the disruption of the macrophage-lymphatic SCS lining by the growing metastasis is part of the colonisation process. This point was not directly addressed in the Response letter, but some additional text has been added to tone down their claims. But directly addressing this concern in the response would be more informative.

We regularly observed that with an increase in the size of metastatic foci in SCS (before invading the LN cortex), a depression was formed by the foci in the SCS macrophage–lymphatic endothelial cell lining. (Figure 1C). We also frequently observed a bump in the LN surface where foci were formed. This suggests that the SCS macrophage–lymphatic endothelial cell lining is flexible and keeps the foci in the SCS sinus until it increases in size. Similarly, Das et al., 2013, observed that there was no disruption of the lymphatic lining of SCS when metastatic foci were present within the sinus. When the foci were big enough to invade, we observed that the normal arrangement of the “SCS macrophage–lymphatic endothelial cell lining” was lost at the foci front where cells enter the LN cortex (Figure 1D). We could see gaps in the lymphatic lining at this stage and some of the frontline metastatic cells invading the LN cortex were still attached to the Siglec1^+^ macrophages that had lost their place in the SCS lining. Therefore, we termed this event as a disruption in the SCS macrophage–lymphatic endothelial cell lining.

Major requests:1) Signaling downstream of Siglec1The authors have addressed this concern by performing co-culture experiments of B16F10 cells with the mouse macrophage line J774A.1 to model SCS macrophages. J774A.1 endogenously expresses Siglec1, and siRNA-mediated knockdown of the endogenous protein resulted in increased apoptosis and decreased proliferation of co-cultured B16F10 cells. This complements the prior results obtained using overexpression of Siglec1 in HEK293 cells. The authors should ensure that for figures concerning the siRNA data, the x-axis is labelled as siRNA (Figure 4G, Figure 2—figure supplement 2 F, H, L) to avoid confusion with the HEK293 overexpression data.

We thank the reviewer for these comments. We have labeled the abovementioned figures to reflect the use of siRNA.

2) The effects of volasertibThe removal of this experiment does help streamline the manuscript and retain focus on Siglec1-mediated downstream signaling.

We thank the reviewer for this comment.

3) Cell-intrinsic properties of Siglec1The experiments performed confirm that altering st3gal3 expression does not have any intrinsic effects on B16F10 cell behaviour, and validated that abrogating siglec1-mediated signaling impairs proliferation and survival of metastasising cells in the SCS.The lack of staining for Siglec in tumour lymphatics shown in the response letter only is somewhat superficial and does not appear to be mentioned in the new document text.

We have added the figure as Figure 5—figure supplement 2 and the following text description to the revised manuscript:

“To explore the possibility of Siglec1 expression on primary tumor lymphatic vessels, which might have affected the initial egress of cancer cells from the primary tumor, we searched for Siglec1 expression on primary tumor lymphatic vessels and found none (Figure 5—figure supplement 2).”

4) Specificity of Siglec1 on macrophagesIt would have been useful to see direct co-staining for Siglec1 and a macrophage marker. There does appear to be faint Siglec staining on LYVE1+ LECs in the medullary sinuses of macrophage-depleted LNs in Figure 1—figure supplement 2B; however the new experiments performed do indicate that the vast majority of Siglec1 in LNs is expressed on sinusoidal macrophages. The results text should describe the macrophage depletion experiment – currently this appears not to be specifically mentioned.

We have added the following text to the revised manuscript:

“Of note, by depleting LN resident macrophages through injecting clodronate liposomes into the mouse footpads, we confirmed that Siglec1 is primarily expressed on SCS and medullary sinus macrophages in LNs (Figure 1—figure supplement 2).”